# Object-centric Learning with Cyclic Walks between Parts and Whole

**Ziyu Wang**[1,2,3]    **Mike Zheng Shou**[1,†]    **Mengmi Zhang**[2,3,†]

[1]Show Lab, National University of Singapore, Singapore

[2]Deep NeuroCognition Lab, CFAR and I2R, Agency for Science, Technology and Research, Singapore

[3]Nanyang Technological University, Singapore

†Corresponding authors; address correspondence to `mengmi@i2r.a-star.edu.sg`

## Abstract

Learning object-centric representations from complex natural environments enables both humans and machines with reasoning abilities from low-level perceptual features. To capture compositional entities of the scene, we proposed cyclic walks between perceptual features extracted from vision transformers and object entities. First, a slot-attention module interfaces with these perceptual features and produces a finite set of slot representations. These slots can bind to any object entities in the scene via inter-slot competitions for attention. Next, we establish entity-feature correspondence with cyclic walks along high transition probability based on the pairwise similarity between perceptual features (aka "parts") and slot-binded object representations (aka "whole"). The whole is greater than its parts and the parts constitute the whole. The part-whole interactions form cycle consistencies, as supervisory signals, to train the slot-attention module. Our rigorous experiments on *seven* image datasets in *three unsupervised* tasks demonstrate that the networks trained with our cyclic walks can disentangle foregrounds and backgrounds, discover objects, and segment semantic objects in complex scenes. In contrast to object-centric models attached with a decoder for the pixel-level or feature-level reconstructions, our cyclic walks provide strong learning signals, avoiding computation overheads and enhancing memory efficiency. Our source code and data are available at: link.

## 1 Introduction

Object-centric representation learning refers to the ability to decompose the complex natural scene into multiple object entities and establish the relationships among these objects [29, 24, 14, 38, 39, 37, 22]. It is important in multiple applications, such as visual perception, scene understanding, reasoning, and human-object interaction [45, 2]. However, learning to extract object-centric representations from complex natural scenes in an unsupervised manner remains a challenge in machine vision.

Recent works attempt to overcome this challenge by relying on image or feature reconstructions from object-centric representations as supervision signals [29, 38, 22, 37] (Figure 1). However, these reconstructions have several caveats. First, these methods often require an additional decoder network, resulting in computation overheads and memory inefficiency. Moreover, these methods focus excessively on reconstructing unnecessary details at the pixel or feature levels and sometimes fail to capture object-centric representations from a holistic view.

To mitigate issues from reconstructions, some studies [47, 19] introduce mutual information maximization between predicted object-centric representations and feature maps. Building upon this idea, contrastive learning has become a powerful tool in unsupervised object-centric learning. The recent works [21, 5] propose to learn the spatial-temporal correspondences between a sequence of video frames with contrastive random walks. The objective is to maximize the likelihood of returning to

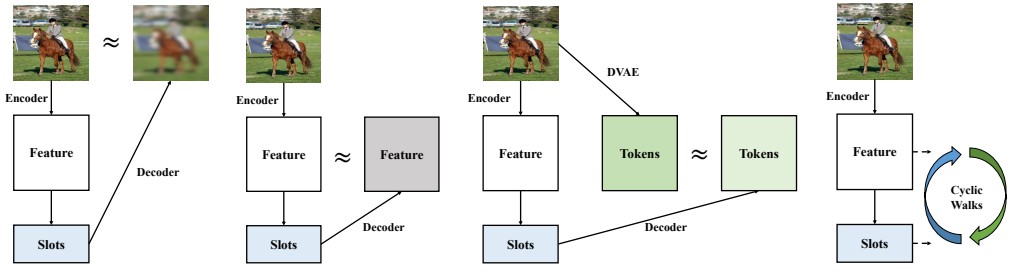

| (a) SA, BO-QSA | (b) DINOSAUR | (c) SLATE | (d) Cyclic walks |
|---|---|---|---|

Figure 1: **Comparison of the network structures trained with existing methods and our cyclic walks.** Supervision signal is indicated by ($\approx$) in subplots a-c. From left to right, all the methods require some forms of reconstructions as supervisory signals except for our method (d): (a) image reconstruction for the Slot-Attention (SA) [29] and BO-QSA [22] (b) feature reconstruction for DINOSAUR [37](c) tokens obtained by a Discrete Variational Autoencoder [36] (DVAE) for SLATE [38] (d). Our cyclic walks do not require any image or feature reconstructions.

the starting node by walking along the graph constructed from a palindrome of frames and repelling nodes with distinct features from adjacent frames in the graph.

Generalizing from contrastive random walks on video sequences, we proposed cyclic walks on static images. These walks are conducted between predicted object entities and feature maps extracted from vision transformers. The cycle consistency of these walks serves as supervision signals to guide object-centric representation learning. First, inspired by DETR [8] and Mask2Former [11], where these methods decompose the scene into multiple object representations or semantic masks with a span of a finite set of task-dependent semantically meaningful bases, we use a slot-attention module to interface with feature maps of an image and produce a set of object-centric representations out of the slot bases. These slot bases compete with one another to bind with feature maps at any spatial location via normalizing attention values over all the slot bases. All the features resembling a particular slot basis are aggregated into one object-centric representation explaining parts of the image.

Next, two types of cyclic walks, along high transition probability based on the pairwise similarity between aggregated object-centric representations and image feature maps, are established to learn entity-feature correspondence. The motivation of the cyclic walks draws a similar analogy with the part-whole theory stating that the whole (i.e. object-centric representations) is greater than its parts (i.e. feature maps extracted from natural images) and the parts constitute the whole. We use "parts" and "whole" for easy illustrations of the interaction between image feature maps and object-centric representations. On one hand, cyclic walks in the direction from the whole to the parts and back to the whole (short for "W-P-W walks") encourage diversity of the learnt slot bases. On the other hand, cyclic walks in the direction from the parts to the whole and back to the parts (short for "P-W-P walks") broaden the coverage of the slot bases pertaining to the image content. Both W-P-W walks and P-W-P walks form cycle consistencies and serve as supervision signals to enhance the specificity and diversity of learned object-centric representations. Our contributions are highlighted below:

- We introduce cyclic walks between parts and whole to regularize object-centric learning. Both W-P-W walks and P-W-P walks form virtuous cycles to enhance the specificity and diversity of the learned object-centric representations.

- We verify the effectiveness of our method over seven image datasets in three unsupervised vision tasks. Our method surpasses the state-of-the-art by a large margin.

- Compared with the previous object-centric methods relying on reconstruction losses, our method does not require additional decoders in the architectures, greatly reducing the computation overheads and improving memory efficiency.

## 2 Related Works

One representative work [29] introduces the slot-attention module, where the slots compete to bind with certain regions of the image via a cross-attention mechanism and distill image features into object-centric representations. To train these slots, the networks are often attached to a decoder to

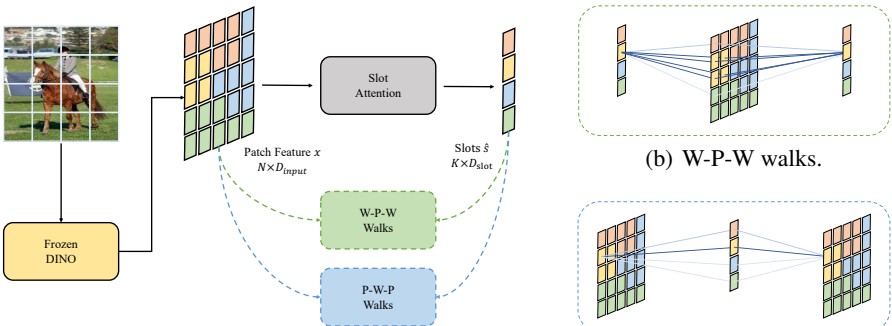

(a) Network architecture of our cyclic walks.        (c) P-W-P walks.

Figure 2: **Overview of our proposed cyclic walks.** The input image is divided into non-overlapped patches and encoded into feature vectors (aka "parts") through a frozen DINO ViT pre-trained on ImageNet with unsupervised learning methods [9]. The Slot-Attention module aggregates the patch features into object-centric representations (aka "whole") based on the learnable slot bases. To train the network, we perform the contrastive random walks in two directions between the parts and the whole: (b) any node from the whole should walk to the parts and back to itself (W-P-W walks) and (c) any node from the parts should walk to the whole and back to itself (P-W-P walks). See Fig A7 for applications of these learned object-centric representations in downstream tasks.

reconstruct either images or features based on object-centric representations (Figure 1). Subsequent Slot-Attention methods [6, 39, 24, 14, 4, 13] expand to video processing. To learn object-level representations, these methods often require either reconstructions from video frames or optical flows [4, 13]. Although the original slot attention was demonstrated to be effective in most cases, it fails to generalize to complex scenes [37, 22]. To mitigate this issue, DINOSAUR *freezes* the feature extractor while learning the slot bases. Its frozen feature extractor was pre-trained on ImageNet in unsupervised learning [9]. BO-QSA [22] proposes to initialize the slots with learnable queries and optimize these queries with bi-level optimization. Both works emphasize that good initialization of network parameters is essential for object-centric learning in complex scenes. Thus, following this line of research, we also use a frozen DINO [9] as the feature extractor. However, different from these works, we introduce cyclic walks in the part-whole hierarchy of the same image, without decoders for reconstructions.

**Object-centric learning for unsupervised semantic segmentation.** One of the key applications in object-centric representation learning is unsupervised semantic segmentation. Several notable works include STEGO [18], DeepCut [1], ACSeg [26], and FreeSOLO [42]. All these methods employ a fixed pre-trained transformer as a feature extractor, which has proven to be effective for unsupervised semantic segmentation. In contrast to these methods which directly perform optimization on pixel clusters or feature clusters, we introduce slot-attention and perform cyclic walks between slots and feature maps. Experimental results demonstrate the superior performance of our method.

**Relation to contrastive learning.** Our method connects object-centric representation learning with unsupervised contrastive learning. We highlight the differences between our method and the contrastive learning methods. Infoseg [19] reconstructs an image from a set of global vectors (aka "slot bases"). The reconstructed image is pulled closer to the original image and repelled away from a randomly selected image. SlotCon [44] introduces contrastive learning on two sets of slot bases extracted from two augmented views of the same image. In contrast to the two methods, our cyclic walks curate both positive and negative pairs along the part-whole hierarchy from the *same* image. Another method, SAN [47], performs contrastive learning between slot bases to encourage slot diversity, where every slot is repelled away from one another. Sharing similar motivations, our walks from the whole to the parts and back to the whole ("W-P-W" walks) strengthen the diversity of the slot bases. Moreover, our method also introduces "P-W-P" walks, which takes the hollistic view of the entire image, and broadens the slot coverage.

## 3 Preliminary

Here, we mathematically formulate **Slot-Attention Module** and **Contrastive Random Walks**.

### 3.1 Slot-Attention Module

The slot-attention module takes an input image and outputs object-centric representations. Given an input image, a pre-trained CNN or transformer extracts its feature vectors $x \in R^{N \times D_{input}}$, where $N = W \times H$. $H$ and $W$ denote the height and width of the feature maps. Taking $x$ and a set of $K$ slot bases $s \in R^{K \times D_{slot}}$ as inputs, Slot-Attention binds $s$ with $x$, and outputs a set of $K$ object-centric feature vectors $\tilde{s}$ of the same size as $s$. The Slot-Attention module employs the cross-attention mechanism [40], where $x$ contributes to keys and values and $s$ contributes to queries. Specifically, the inputs $x$ are mapped to the dimension $D$ with linear functions $k(\cdot)$ and $v(\cdot)$ and the slots $s$ are mapped to the same dimension with linear function $q(\cdot)$. For each feature vector at every spatial location of the feature maps $x$, attention values are calculated with respect to all slot bases. To prevent parts of the image from being unattended, the attention matrix is normalized with the softmax function first over K slots and then over $N$ locations:

$$\texttt{attn}_{i,j} = \frac{e^{A_{i,j}}}{\sum\limits_{K} e^{A_{i,j}}} \text{ where } A = \frac{k(x)q(s)^{\mathsf{T}}}{\sqrt{D}} \in R^{N \times K} \tag{1}$$

$$\tilde{s} = W^T v(x) \text{ where } W_{i,j} = \frac{\texttt{attn}_{i,j}}{\sum\limits_{N} \texttt{attn}_{i,j}} \tag{2}$$

The cross-attention mechanism introduces competitions among slot bases for explaining parts of the image. Based on the attention matrix, an object-centric feature vector from $\tilde{s}$ is distilled by the weighted sum of feature maps $x$. In the literature [8, 38, 29] and our method, the slot attention modules iteratively refine the output $\tilde{s}$ from the slots in a recurrent manner with a Gated Recurrent Unit (GRU) for better performances. We represent the initial slot bases as $s^0$. The parameters of $s^0$ are initialized by randomly sampling from Gaussian distribution with learnable mean $\mu$ and variance $\sigma$. The output $\tilde{s}$ at time step $t$ acts as the slot bases and feeds back to the GRU at the next recurrent step $t + 1$. Thus, the final object-centric representations $\hat{s}$ after $T$ iterations can be formulated as:

$$\hat{s} = s^T, \text{ where } s^{t+1} = GRU(s^t, \tilde{s}^t) \tag{3}$$

After obtaining the object-centric representation $\hat{s}$, the traditional object-centric learning models decode the images from the slots with a mixture-based decoder or a transformer-based decoder [8]. The training objective of the models is to minimize the Mean Square Error loss between the output of the decoder and the original image at the feature or pixel levels.

### 3.2 Contrastive Random Walks

A random walk describes a random process where an independent path consists of a series of hops between nodes on a directed graph in a latent space. Without loss of generality, given any pair of feature sets $a \in R^{m \times d}$ and $b \in R^{n \times d}$, where $m$ and $n$ are the numbers of nodes in the feature sets and $d$ is the feature dimension, the adjacency matrix $M_{a,b}$ between $a$ and $b$ can be calculated as their normalized pairwise feature similarities:

$$M_{a,b} = \frac{e^{f(a)f(b)^T/\tau}}{\sum\limits_{n} e^{f(a)f(b)^T/\tau}} \in R^{m \times n}, \tag{4}$$

where $f(\cdot)$ is the $l_2$-normalization and $\tau$ is the temperature controlling the sharpness of distribution with its smaller values indicating sharper distribution.

In the original work [21], random walks serve as supervisory signals to train the object-centric learning models to capture the space-time correspondences from raw videos. The contrastive random walks are formulated as a graph. $a$ and $b$ are the feature maps extracted from video frames with either a CNN or a transformer. $m$ and $n$ are the numbers of spatial locations on the feature maps and they are often the same across video frames. On the graph, only nodes from adjacent video frames $F_t$ and $F_{t+1}$ at time $t$ and $t+1$ share a directed edge. Their edge strengths, indicating the transition probabilities of a random walk between frames, are the adjacency matrix $M_{F_t, F_{t+1}}$. The objective of the contrastive random walks is to maximize the likelihood of a random walk returning to the starting node along the graph constructed from a palindrome video sequence $\{F_t, F_{t+1}, ..., F_T, F_{T-1}, ..., F_t\}$.

## 4 Our Method

### 4.1 Object-centric Feature Encoding

Following [18, 37], we use a self-supervised vision transformer trained with DINO on ImageNet, as the image feature extractor (Figure 2a). In Section 5.4, we verify that our method is also agnostic to other self-supervised feature learning backbones. Given an input image, DINO parses it into non-overlapped patches and each patch is projected into a feature token. We keep all patch tokens except for the classification token in the last block of DINO. As in Section 3.1, we use the same notation $x \in R^{N \times D_{input}}$ to denote feature maps extracted from a static image. The Slot-Attention module takes the feature vectors $x \in R^{N \times D_{input}}$ and a set of K learnable slot bases $s \in R^{K \times D_{slot}}$ as inputs and produces the object-centric representations $\hat{s} \in R^{K \times D_{slot}}$ ( Equations 1 and 2).

### 4.2 Whole-Parts-Whole Cyclic Walks

Features of the image patches constitute the objects. The interactions between parts and the whole provide a mechanism for clustering and discovering objects in the scene. Motivated by this, we introduce cyclic walks in two directions: (a) from the whole to the parts and back to the whole (W-P-W walks) and (b) from the parts to the whole and back to the parts (P-W-P walks). Both serve as supervisory signals for learning slot bases $s$.

Given feature vectors $x \in R^{N \times D_{input}}$ of all image patches (aka "parts") and the object-centric representations $\hat{s} \in R^{K \times D_{slot}}$ (aka "whole"), we apply a linear layer to map both $x$ and $\hat{s}$ to be of the same dimension $D$. In practice, K is much smaller than N. Following the formulations of Contrastive Random Walks [21] in Section 3.2, random walks are conducted along high transition probability based on pairwise similarity from $\hat{s}$ to $x$ and from $x$ to $\hat{s}$ using Equation 4:

$$M_{wpw} = M_{\hat{s},x} M_{x,\hat{s}} \in R^{K \times K}, \text{ where } M_{\hat{s},x} \in R^{K \times N} \text{ and } M_{x,\hat{s}} \in R^{N \times K} \tag{5}$$

$M$ is defined as an adjacency matrix. Different from the original work [21] performing contrastive random walks on a palindrome video sequence, here, we enforce a palindrome walk in a part-whole hierarchy of the image. Ideally, if the W-P-W walks are successful, all the bases in the slot-attention module have to establish one-to-one mapping with certain parts of the image. This encourages the network to learn diverse object-centric representations so that each slot basis can explain certain parts of the image and every part of the image can correspond to a unique slot basis. To achieve this in W-P-W walks, we enforce that the probability of walking from $\hat{s}$ to $x$ belonging to the object class and back to $\hat{s}$ itself should be an identity matrix $I$ of size $K \times K$. The first loss term is defined as: $L_{wpw} = CE(M_{wpw}, I)$, where $CE(\cdot, \cdot)$ refers to cross entropy loss.

### 4.3 Parts-Whole-Parts Cyclic Walks

Though the W-P-W walks enhance the diversity of the slot bases, there is an ill-posed situation, when there exists a limited set of slot bases failing to cover all the semantic content of an image but every trivial W-P-W walk is always successful. For example, given two slot bases and an image consisting of a triangle, a circle, and a square on a background, one slot could prefer "triangles", while the other prefers "circles". In this case, the W-P-W walks are always successful; however, the two slots fail to represent squares and the background, defeating the original intention of foreground and background extraction. To mitigate this problem and bind all the slots with all the semantic content in the entire scene, we introduce additional walks from the parts to the whole and back to the parts (P-W-P walks), complementary to W-P-W walks: $M_{pwp} = M_{x,\hat{s}} M_{\hat{s},x} \in R^{N \times N}$. As N is typically far larger than K and features $x$ at nearby locations tend to be similar, the probabilities of random walks beginning from one feature vector of $x$, passing through $\hat{s}$, and returning to itself could no longer be 1. Thus, we use the feature-feature correspondence $S$ as the supervisory signal to regularize $\hat{s}$:

$$S_{i,j} = \frac{e^{W_{i,j}}}{\sum_{N} e^{W_{i,j}}} \text{ where } W_{i,j} = \begin{cases} -\infty, & \text{if } F_{i,j} <= \gamma \\ F_{i,j}, & \text{if } F_{i,j} > \gamma \end{cases}, \text{ and } F = f(x)f(x)^T \in R^{N \times N} \tag{6}$$

Hyper-parameter $\gamma$ is the similarity threshold for preventing the random walks from returning to locations where their features are not as similar as the starting point. Thus, the overall loss of our

| Model | Birds | | Dogs | | Cars | | Flowers | |
|---|---|---|---|---|---|---|---|---|
| | mIoU | Dice | mIoU | Dice | mIoU | Dice | mIoU | Dice |
| Slot-Attention [29] | 35.6 | 51.5 | 39.6 | 55.3 | 41.3 | 58.3 | 30.8 | 45.9 |
| SLATE [38] | 36.1 | 51.0 | 62.3 | 76.3 | 75.5 | 85.9 | 68.1 | 79.1 |
| DINOSAUR [37] | 67.2 | 78.4 | 73.7 | 84.6 | 80.1 | 87.6 | 72.9 | 82.4 |
| BO-QSA [22] | 71.0 | 82.6 | 82.5 | 90.3 | 87.5 | 93.2 | **78.4** | **86.1** |
| Cyclic walks (ours) | **72.4** | **83.6** | **86.2** | **92.4** | **90.2** | **94.7** | 75.1 | 83.9 |

Table 1: **Results in the unsupervised foreground extraction task.** We report the results in mIoU and Dice on CUB200 Birds (Birds), Stanford Dogs (Dogs), Stanford Cars (Cars), and Flowers (Flowers) datasets. Numbers in bold are the best.

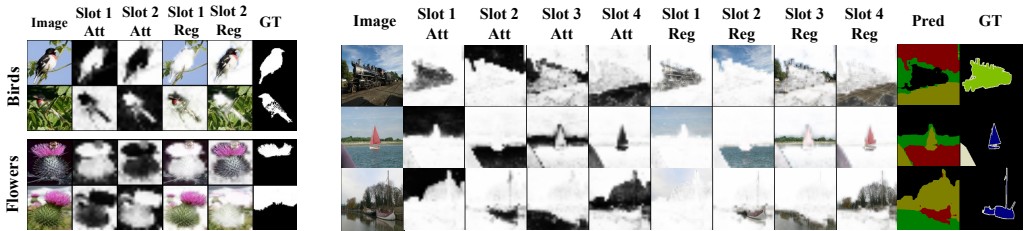

(a) Foreground extraction          (b) Object discovery

Figure 3: **Visualization of attention maps predicted by slot bases.** In (a) unsupervised foreground extraction, we present two examples each from Birds, and Flowers datasets. In each example, for all the slots, we provide their attention maps (Col.2-3) and their corresponding attended image regions (Col.4-5). Ground truth foreground and background masks are shown in the last Col. In (b) unsupervised object discovery, we provide 3 examples on the PASCAL VOC dataset. In each example, for all the slots, we provide their attention maps (att., Col.2-5) and their corresponding attended image regions (reg., Col. 6-9). Their combined object discovery masks from all the slots are shown in Col.10 (pred.). Each randomly assigned color denotes an image region activated by one slot. Ground truth object masks are presented in the last column.

cyclic walks can be calculated below, where $\alpha$ and $\beta$ are the coefficients balancing the two losses.

$$L = \alpha L_{wpw} + \beta L_{pwp}, \text{ where } L_{pwp} = CE(M_{pwp}, S) \tag{7}$$

**Training Configuration.** We use ViT-Small pre-trained with DINO as our feature extractor and freeze the parameters of DINO throughout the entire training process. There are multiple reasons for freezing the feature extractor. First, pre-trained unsupervised feature learning frameworks have already generated semantically consistent content in nearby locations [18, 1, 26]. Object-centric learning addresses its follow-up problem of capturing compositional entities out of the semantic content. Second, recent object-centric learning works [37, 22] have emphasized the importance of freezing the pre-trained weights of feature extractors (see Section 2). For a fair comparison with these methods, we keep the parameters of the feature extractor frozen. Third, we provided empirical evidence that our method benefits from freezing DINO, and freezing DINO is not a limitation of our method in **Appendix A1**. We use a similarity threshold of 0.7 and a temperature of 0.1 for all the experiments over all the datasets. In general, the variations of these two hyperparameters lead to moderate changes in performances (see Sec 5.4). See **Appendix A2** for more training and implementation details. All models are trained on 4 Nvidia RTX A5000 GPUs with a total batch size of 128. We report the mean ± standard deviation of 5 runs with 5 random seeds for all our experiments.

## 5 Experiments

### 5.1 Unsupervised Foreground Extraction

**Task Setting, Metrics, Baselines and Datasets.** In the task, all models learn to output binary masks separating foregrounds and backgrounds in an unsupervised manner. See **Appendix A3.1** for implementations of foreground and background mask predictions during the inference. We evaluate the quality of the predicted foreground and background masks with mean Intersection over Union

(mIoU) [22] and Dice [22]. The mIoU measures the overlap between predicted masks and the ground truth. Dice is similar to mIoU but replaces the union of two sets with the sum of the number of their elements and doubles the intersection. We used publicly available implementations of Slot-Attention [29] and SLATE [38] from [22] and re-implemented DINOSAUR [37] by ourselves. Note that the results of re-implemented DINOSAUR deviate slightly from the original results in [37]. We provided further comparisons and discussed such discrepancy in **Appendix A4**. However, none of these alter the conclusions of this paper. Following the work of BO-QSA [22], we include Stanford Dogs [23], Stanford Cars [25], CUB 200 Birds [43], and Flowers [32] as benchmark datasets.

**Results and Analysis.** The results in mIoU and Dice are presented in Table 1. Our method achieved the best performance on the Birds, Dogs, and Cars datasets and performed competitively well as BO-QSA on the Flowers dataset. Slot-attention, SLATE, and DINOSAUR use the pixel-level, feature-level, and token-level reconstruction losses from object-centric representations as supervisory signals. The inferior performance of slot attention over SLATE and DINOSAUR implies that pixel-level reconstructions focus excessively on unnecessary details, impairing the object-centric learning performance. We also note that DINOSAUR outperforms Slot-Attention and SLATE by a significant margin. This aligns with the previous findings that freezing pre-trained feature extractors facilitates object-centric learning. In addition to reconstructions, BO-QSA imposes extra constraints on learnable slot queries with bi-level optimizations, which leads to better performances. This emphasizes the necessity of searching for better supervisory signals to regularize object-centric learning beyond reconstructions. Indeed, even without any forms of reconstruction, we introduce cyclic walks between parts and whole as an effective training loss for object-centric learning. Our method yields the best performance among all comparative methods. The result analyses hold true in subsequent tasks (Section 5.2 and 5.3).

We also visualize the attention masks predicted by our slot bases in Figure 3(a) (see **Appendix A5.1** for more visualization results). We observed that the model trained with our cyclic walks outputs reasonable predictions. However, we also noticed several inconsistent predictions. For example, our predicted masks in the Flowers dataset (Row 3 and 4) group the pedals and the sepals of a flower altogether, because these two parts often co-occur in a scene. Moreover, we also spotted several wrongly annotated ground truth masks. For example, in Row 4, the foreground mask mistakenly incorporates the sky as part of the ground truth. It is also interesting to see that the foreground object belonging to the same object class is not always identified by a fixed slot basis (compare Row 1 and 2 in the Birds dataset). During the inference, we randomly sample parameters from learnable mean $\mu$ and variance $\sigma$ as the slot bases (Section 3.1), which could cause the reverse of foreground and background masks. Applying the same set of parameters for slot bases across the experiments avoids such issues.

## 5.2 Unsupervised Object Discovery

**Task Setting, Metrics, Baselines and Datasets.** The task aims to segment the image with a set of binary masks, each of which covers similar parts of the image, (aka "object masks"). However, different from image segmentation, each of the masks does not have to be assigned specific class labels. We follow recent works and evaluate all models [29, 38, 22, 37] and K-Means with ARI on foreground objects (ARI-FG). ARI reflects the proportion of sampled pixel pairs on an image, correctly classified into the same class or different classes. All baseline implementations are based on the open-source codebase [28]. Same as Section 5.1, we use the $M_{x,\hat{s}}$ as the object masks. $M_{x,\hat{s}}$ is the similarity matrix, indicating the transition probability between features and slots (see Equation 4 and Equation 5). We benchmark all methods on the common datasets Pascal VOC 2012 [15], COCO 2017 [27], Movi-C [17] and Movi-E[17]. We also evaluate all methods on a synthetic dataset ClevrTex for further comparisons (see **Appendix A6**).

**Results and Analysis** The results in ARI-FG on the four datasets are shown in Figure 4. The unsupervised object discovery task is harder than the unsupervised foreground extraction task (Section 5.1) due to the myriad scene complexity and object class diversity. Our method still consistently outperforms all SOTA methods and beats the second-best method DINOSAUR by a large margin of 2 - 4% over all four datasets. Different from all other methods attached with decoders for the image or feature reconstructions, our model trained with cyclic walks learns better object-centric representations. We visualized some example predicted object masks in Figure 3(b) (see **Appendix A5.2** for more positive samples). Our method can clearly distinguish semantic areas in complex

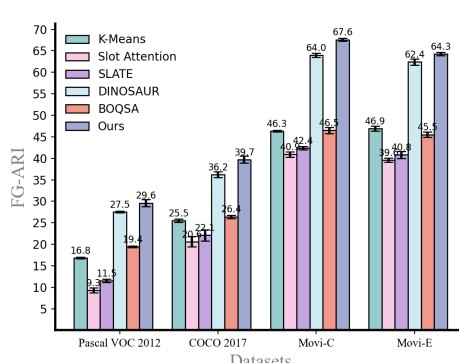

Figure 4: **Performance of Object Discovery.** From left to right, the used datasets are Pascal VOC 2012 [15], COCO 2017 [27], Movi-C [17] and Movi-E [17]. We report the performance in foreground adjusted rand index (ARI-FG). The higher the ARI-FG, the better. We compare our method (cyclic walks) with K-means, Slot-Attention [29], SLATE [38], DINOSAUR [37], and BO-QSA [22].

| Model | Pascal VOC 2012 | COCO Stuff-27 | COCO Stuff-3 |
|---|---|---|---|
| ACSeg [26] | 47.1 | 16.4 | - |
| MaskDistill [16] | 42.0 | - | - |
| PiCIE + H [12] | - | 14.4 | - |
| STEGO [18] | - | **26.8** | - |
| SAN [47] | - | - | 80.3 |
| InfoSeg [19] | - | - | 73.8 |
| SlotCon [44] | - | 18.3 | - |
| COMUS [46] | **50.0** | - | - |
| Cyclic walks (ours) | 43.3 ± 1.6 | 22.5 ± 1.1 | **82.4 ± 0.7** |

Table 2: **Results of Unsupervised Semantic Segmentation in IoU on Pascal VOC 2012, COCO-Stuff-27 [7] and COCO-Stuff-3 [7].** (±) standard deviations of IoU over 5 runs are also reported. The '-' indicates that corresponding results are not provided in the original papers and the source codes are unavailable. Best is in bold.

scenes. For example, trains, trees, lands, and sky are segmented in Row 1 in an unsupervised manner. Note that the foreground train masks are provided as the ground truth annotations in PASCAL VOC only for qualitative comparisons. They have never been used for training. Additional visualization results on MOVi can be found in **Appendix A5.3**. In the experiments, we also noticed several failure cases. When the number of semantic classes in the image is less than the number of slots, our method segments the edges of semantic classes (see **Appendix A5.2** for more analysis).

### 5.3 Unsupervised Semantic Segmentation

**Task Setting, Metrics, Baselines and Datasets.** In the task, each pixel of an image has to be classified into one of the pre-defined object categories. See **Appendix A3.2** for implementations of obtaining the category labels for each predicted mask during inference. We report the intersection over union (IoU) between the predicted masks and the ground truth over all categories. We include ACSeg [26], MaskDistill [16], PiCIE [12], STEGO [18], SAN [47], InfoSeg [19], SlotCon [44], and COMUS [46] for comparison. All the results are directly obtained from their original papers. We evaluate all competitive methods on Pascal VOC 2012, COCO Stuff-27 [7], and COCO Stuff-3 [7]. COCO Stuff-27 and COCO Stuff-3 contain 27 and 3 supercategories respectively.

**Results and Analysis** We report the results in terms of IoU in Table 2. Our method has achieved the best performance on COCO-Stuff-3 and the second best on Pascal VOC 2012 and COCO-Stuff-27. This highlights that our cyclic walks in the part-whole hierarchy on the same images act as effective supervision signals and distill image pixels into diverse object-centric representations rich in semantic information. It is worth pointing out that our model is not specially designed for semantic segmentation. Yet, it is remarkable that our method still performs competitively well as the majority of the existing semantic segmentation models.

On Pascal VOC 20212, our method slightly underperforms COMUS [46] in mIoU metrics (43.3% versus 50.0%). We argue that, in comparison to our method, there are two additional components in COMUS possibly attributing to the performance differences and unfair comparisons. First, COMUS employs two pre-trained saliency detection architectures DeepUSPS [31] and BasNet [35] in addition to DINO. The saliency detection model requires an additional MSRA image dataset [34] during pre-training. Thus, compared to our cyclic walks, COMUS indirectly relies on extra useful information. Second, COMUS uses Deeplab-v3 as its backbone for predicting semantic segmentation masks. For better segmentation performances, Deeplab-v3 extracts large-resolution feature maps from images

| Dataset | Full Model | Random Walk Direction | | Temperature of Random Walks | | | Number of slots | | | Similarity Threshold | | | Feature Extractor | | |
|---|---|---|---|---|---|---|---|---|---|---|---|---|---|---|---|
| | | P-W-P | W-P-W | 0.07 | 0.3 | 1 | 5 | 6 | 7 | -inf | 0.3 | 1 | Moco-v3 [10] | MAE [20] | MSN [3] |
| Pascal VOC 2012 | **29.6** | 27.1 | 28.4 | 23.5 | 27.3 | 21.9 | 27.7 | 26.2 | 24.3 | 22.5 | 26.5 | 26.2 | 29.3 | 26.8 | 29.0 |
| | | P-W-P | W-P-W | 0.07 | 0.3 | 1 | 10 | 12 | 13 | -inf | 0.3 | 1 | Moco-v3 [10] | MAE [20] | MSN [3] |
| COCO 2017 | **39.7** | 34.8 | 36.4 | 35.5 | 36.7 | 29.6 | 37.9 | 35.7 | 35.3 | 29.8 | 32.3 | 31.8 | 38.4 | 34.5 | 38.2 |

Table 3: **Ablation Studies and Method Analysis on Pascal VOC 2012 and COCO 2017 in terms of ARI-FG.** Full model refers to our default method introduced in Section 4. See **Appendix A2** for the empirically determined hyper-parameter details of our full model. We vary one factor at a time and study its effect on ARI-FG performances in the Pascal VOC 2012 (Row 3) and COCO 2017 (Row 5). The best is our default model highlighted in bold. See Section 5.4 for details.

and applies Atrous Spatial Pyramid Pooling for aggregating these feature maps over multiple scales. In contrast, our method extracts low-resolution feature maps with DINO and is not specifically designed for unsupervised semantic segmentation.

Despite a slightly inferior performance to COMUS, our method is capable of parsing the entire scene on the image into semantically meaningful regions, which COMUS fails to do. These include distinct backgrounds, such as sky, lake, trees, and grass. For example, in Fig 3(b), our method successfully segments backgrounds, such as the lake, the trees, and the sky. However, COMUS fails to segment background elements. For example, in Column 4 of Fig 1 in the paper of COMUS [46], COMUS only segments the foreground ships and fails to segment the lake and the sky. Segmenting both salient objects and other semantic regions is essential for many computer vision applications. This emphasizes the importance and unique advantages of our method. We also attempted to apply object-centric methods in instance segmentation tasks. In the **Appendix A5.4**, we provided visualization results and discussions.

## 5.4 Ablation Study and Method Analysis

We examine the effect of hyperparameters, the components of training losses, and the variations of feature extractors on object-centric learning performances and report ARI-FG scores for all these experiments on Pascal VOC 2012 [15] and COCO 2017 [27] (Table 3).

**Ablation on Random Walk Directions.** To explore the effects of random walk directions, we use either P-W-P or W-P-W walks to train the network separately and observe the changes (Table 3, Col.3-4). We found that either direction of random walks can make the training loss of the network converge. The W-P-W walks perform slightly better than the P-W-P walks, but neither of these walks surpasses random walks in both directions. This aligns with the design motivation of our method (Section 4).

**Analysis on Temperature of Random Walks.** We titrate the temperate in Equation 4 from 0.07 to 1 and observe a non-monotonic performance trend versus temperature choices (Col.5-7). On one hand, the lower the temperature, the more concentrated the attention distribution over all the slots and the faster the network can converge due to the steeper gradients. On the other hand, the attention distribution becomes too sharp with much lower temperatures, resulting in optimization instability.

**Analysis on Number of Slots.** The number of slots imposes constraints on the diversity of objects captured in the scene. We trained models with various numbers of slots (Col.8-10). More slots do not always lead to better performances. With more slots, it is possible that all slots compete with one another for each image patch and the extra slots only capture redundant information, hurting the object discovery performance.

**Analysis on Similarity Threshold in P-W-P Walks.** During P-W-P cyclic walks, we introduce similarity threshold $t$ (Equation 6). From Col.11-13, we can see that with the increase of threshold $\gamma$ from $-inf$ to 0.7, the P-W-P random walks become more selective, enforcing the slot bases to learn more discriminative features; hence, better performance in object discovery tasks. However, when the threshold approaches 1, the ability to walk to neighboring image patches with high semantic similarity to the starting patch is impaired, leading to the overfitting of slot bases.

**Analysis on Different Feature Extractors.** We replace pre-trained DINO transformer (Section 4.1) with MOCO-V3 [10], MAE [20], and MSN [3]. Together with DINO, these feature extractors are state-of-the-art unsupervised representation learning frameworks. From Col.14-16, we observe that various backbones with our method consistently achieve high ARI-FG scores over both datasets. This suggests that our method is agnostic to backbone variations and it can be readily adapted to any general SOTA unsupervised learning frameworks.

**Analysis on Reconstruction Loss.** We assess the effect of pixel-level reconstructions by adding a transformer-based decoder and introducing an extra reconstruction loss. The upgraded model achieved slightly better performance than our default model (from 29.6% to 29.9% in ARI-FG) on Pascal VOC 2012. This indicates that the reconstruction loss provides auxiliary information for object-centric learning. In future work, we will explore the possibilities of predicting object properties from the slot features.

## 5.5 Efficiency Analysis in Model Sizes, Training Speed, GPU Usages, and Inference Speed

| Method | Parameters $(10^3)$ | Training speed (image / sec) | GPU usage (M) | Inference speed (image / sec) |
|---|---|---|---|---|
| Slot-Attention | 3144 | 114 | 4371 | 126 |
| SLATE | 14035 | 27 | 16327 | 32 |
| DINOSAUR | 11678 | 124 | 3443 | 130 |
| BO-QSA | 14223 | 25 | 16949 | 32 |
| Cyclic walks (ours) | **1927** | **208** | **2331** | **285** |

Table 4: **Method Comparison in the number of parameters, training speed, GPU memory, and inference speed.** From top to bottom, we include Slot-Attention, SLATE, DINOSUAR and BO-QSA. The inference speed was reported based on Object Discovery. The best is in bold.

We report the number of network parameters, training speed, GPU usage, and inference speed in Table 4. All these experiments are run with the same hardware specifications and method configurations: (1) one single RTX-A5000 GPU; and (2) input image size of $224 \times 224$, batch size of 8, and 4 slots. In comparison with all transformer decoder-based methods (SLATE, DINOSAUR and BO-QSA), our method uses the fewest parameters and yet, achieves the best performance in all the tasks (Sections 5.1, 5.2, 5.3). Compared to other decoder-based methods, Slot-Attention requires fewer parameters by sampling from a mixture of slot bases for image reconstruction. However, our method requires even fewer parameters than Slot-Attention (Col. 1). Moreover, although the transformer-based networks are slower in training compared with the CNN-based networks, DINOSAUR, and our method are much faster than SLATE and BO-QSA due to freezing the feature extractor (Col. 2). Besides, the losses of our method converge the fastest. On Pascal VOC 2012 and COCO 2017, our model can achieve the best performance within 10k training steps, while other models need at least 100k training steps. By additionally benefiting from cyclic walks and bypassing decoders for feature reconstruction, our method runs twice as much faster as DINOSAUR. With the same reasoning, the networks trained with our cyclic walks only require half of the GPU memory usage of DINOSAUR and achieve the fastest inference speed without sacrificing the performance in all three tasks (Col. 3 and Col. 4).

## 6 Discussion

We propose cyclic walks in the part-whole hierarchy for unsupervised object-centric representation learning. In the slot-attention module, a finite set of slot bases compete to bind with certain regions of the image and distill into object-centric representations. Both P-W-P and W-P-W cyclic walks serve as implicit supervision signals for training the networks to learn compositional entities of the scene. Subsequently, these entities can be useful for many practical applications, such as scene understanding, reasoning, and explainable AIs. Our experiments demonstrate that our cyclic walks outperform all competitive baselines over seven datasets in three unsupervised tasks while being memory-efficient and computation-efficient during training. So far, our method has been developed on a frozen unsupervised feature extractor. In the future, hierarchical contrastive walks can be explored in any feed-forward architectures, where the models can simultaneously learn both pixel-level and object-centric representations incrementally over multiple layers of the network architecture. We provide in-depth discussion of limitations and future works in **Appendix A7**.

## Acknowledgments and Disclosure of Funding

This research is supported by the National Research Foundation, Singapore under its AI Singapore Programme (AISG Award No: AISG2-RP-2021-025), and its NRFF award NRF-NRFF15-2023-0001. Mike Zheng Shou is supported by the National Research Foundation, Singapore under its NRFF Award NRF-NRFF13-2021-0008. We also acknowledge Mengmi Zhang's Startup Grant from Agency for Science, Technology, and Research (A*STAR), and Early Career Investigatorship from Center for Frontier AI Research (CFAR), A*STAR.

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

## A1 Discussion of Using Pre-trained Features

In this section, we explain why we use the frozen feature extractor and highlight the advantages of using it. In addition, we fine-tune the feature extractor and the inferior results prove the importance of using frozen feature extractors.

### A1.1 Using Pre-trained Features Is NOT a Limitation of Our Method.

We follow the same practice as previous works [37, 16] by freezing feature extractors. It is for a fair comparison with the baselines in the literature.

We argue that freezing feature extractors is not a limitation of our method because of the two reasons below. First, our method can still be easily adapted to any new datasets with domain shifts. To achieve this, we introduce the 2-stage training pipeline. At stage 1, the feature extractor of a model can be trained on the new dataset using self-supervised learning. At stage 2, the feature extractor is frozen and the slot-based attention module can be fine-tuned using our method on the new dataset. Note that both stages only require self-supervised learning without any human supervision. This enables our method to be easily applied in new datasets and new domains with 2-stage training.

Second, freezing feature extractors before learning object-centric representations is loosely connected with the neuroscience findings. These findings suggest that neural plasticity is increased from low-level visual processes (frozen feature extractor) to higher levels of cortical processing responsible for handling symbolic representations (slot-based attention module).

### A1.2 Fine-tuning the Feature Extractor Hurts the Performances.

| Method | FG-ARI |
|---|---|
| 1-Fine-tune | 12.2 |
| 2-Learning-rate | 22.4 |
| 3-EMA | 21.3 |
| Frozen | **29.6** |

Table A1: **Results of different methods for fine-tuning the feature extractor.** We use the experiment settings in the object discovery task on Pascal VOC 2012 and report the FG-ARI.

We did the following three experiments to investigate how fine-tuning the feature extractor contributes to the overall performance of object-centric learning in ARI-FG. The results are shown in Table A1. First, we fine-tune both the feature extractor and the slot-based attention (1-Fine-tune). The performance in ARI-FG is 12.2%. Second, we assign a small learning rate of 0.0001 for the feature extractor and a large learning rate of 0.0004 for the slot-based attention (2-Learning-rate). We observed a great performance improvement from 12.2% in 1-Fine-tune to 22.4% in 2-Learning-rate. Third, we apply EMA (Exponential Moving Averaging) on the entire model (3-EMA). The performance of 21.3% in 3-EMA is slightly worse than the 2-Learning-rate.

From all these results, aligning with the neuroscience inspirations above in the previous subsection, we found that the slow update of the feature extractor stabilizes the learning of high-level object-centric representations. However, the performance is still inferior to our default model in the paper (29.6% in ARI-FG). This emphasizes the importance of freezing feature extractors for our method.

## A2 Training Details and Model Configurations

We set the patch size to be 8. Our model is optimized by AdamW optimizer [30] with a learning rate of 0.0004, 250k training steps, linearly warm-up of 5000 steps, and an exponentially weight-decaying schedule. The gradient norm is clipped at 1. We use Pytorch automatic mixed-precision and data paralleling for training acceleration. All models are trained on 4 Nvidia RTX A5000 GPUs with a total batch size of 128. The temperature of cyclic walks is set to 0.1. We use similarity threshold 0.7 and ViT-S8 of DINO [9] for all experiments. We report the mean $\pm$ standard deviation of 5 runs with 5 random seeds for all our experiments. We trained all models (Slot-Attention, SLATE, BO-QSA, DINOSAUR, and our Cyclic walks) with 250k training steps and selected their best models by keeping track of their best accuracies on the validation sets.

We list the number of slots and image size used for each dataset in Table A1. For Birds, Cars, Dogs, and Flowers datasets, we report the performance of Slot-Attention, SLATE, and BO-QSA from the work BO-QSA [22]. For other implementation details, we follow all method configurations in the work DINOSAUR [37].

| | Birds | Cars | Dogs | Flowers | Pascal VOC 2012 | COCO 2017 | COCO-Stuff | Movi-C | Movi-E |
|---|---|---|---|---|---|---|---|---|---|
| Slot-Attention | - | - | - | - | 6 | 7 | - | 11 | 24 |
| SLATE | - | - | - | - | 6 | 7 | - | 11 | 24 |
| DINOSAUR | 2 | 2 | 2 | 2 | 4 | 7 | - | 11 | 24 |
| BO-QSA | - | - | - | - | 6 | 7 | - | 11 | 24 |
| Cyclic walks (ours) | 2 | 2 | 2 | 2 | 4 | 7 | 11 | 11 | 24 |
| Image Size | 128 | 128 | 128 | 128 | 224 | 224 | 224 | 224 | 224 |

Figure A1: The choice of the number of slots and image size for all the methods in each dataset. The '-' indicates that the performance results are directly taken from other papers and thus the configuration is not provided here.

## A3 Inference Steps of Our Method

### A3.1 Unsupervised Foreground Extraction and Unsupervised Object Discovery

During the inference, all the models are asked to predict foreground masks in the unsupervised foreground extraction task and object masks in the unsupervised object discovery task (Section 5.2). We use $M_{x,\hat{s}}$ (Equation 5) as the segmentation masks, where each feature vector at any spatial location of $x$ is softly assigned to a cluster center. The mask with a maximum intersection with the ground truth masks is viewed as the corresponding predicted mask.

### A3.2 Unsupervised Semantic Segmentation

In the task, each pixel of an image has to be classified into one of the pre-defined object categories. To obtain the category labels for each predicted mask, we perform the following inference steps. (a) we obtain a set of object-centric representations $\hat{s}$ for each image. (b) We compute all the object features by taking matrix multiplication between $M_{\hat{s},x}$ and $x$ from all the images and then perform k-means clustering on these feature vectors, in which the number of clusters is the number of semantic categories of the benchmark. (c) A binary matching algorithm is used to match our clustered categories with the ground truth by maximizing mIoU. (d) Each pixel on a given image can be assigned to the predicted class label corresponding to the binded slot basis.

## A4 Extended Comparison and Discussion with DINOSAUR

| | Pascal VOC 2012 | COCO 2017 | MOVi-C | MOVi-E |
|---|---|---|---|---|
| Original DINOSAUR | $24.6 \pm 0.2$ | $40.5 \pm 0.0$ | $67.2 \pm 0.3$ | $64.7 \pm 0.7$ |
| Re-implemented DINOSAUR | $27.5 \pm 0.2$ | $36.2 \pm 0.7$ | $64.0 \pm 0.5$ | $62.4 \pm 0.7$ |
| Cyclic walks (ours) | $29.6 \pm 0.8$ | $39.7 \pm 0.8$ | $67.6 \pm 0.3$ | $64.3 \pm 0.7$ |

Table A2: **Comparison of the original DINOSAUR and our re-implementation on Object discovery.** We report the foreground adjusted rand index (FG-ARI). ($\pm$) stands for the standard deviations.

Note that the results of DINOSAUR reported in our paper deviate slightly from the results reported in the original DINOSAUR paper [37]. This is because the code for DINOSAUR was not available at the point of submission. Hence, we strictly followed the model specifications in the original paper, re-implemented DINOSAUR on our own, and reported the results of our re-implemented version. These results include the performances of our re-implemented DINOSAUR on Birds, Dogs, Cars, and Flowers datasets, which were missing in the original DINOSAUR paper. We released our re-implementation code of all the baselines at link.

In Table A2, we copied the exact same results in the original DINOSAUR paper. For easy comparison, we also copied the results of our re-implemented DINOSAUR and our method from our paper. The ARI-FG metric performances in (mean+/- standard deviation) are reported. From the results, we observed that our method outperformed the re-implemented DINOSAUR and performed competitively well as the original DINOSAUR in all the experiments. Note that in comparison to DINOSAUR, our

method does not require decoders for reconstruction and hence, our method has fewer parameters, less GPU memory usage, converges faster during training, and achieves faster inference speed as indicated in Sec 5.5.

## A5  Additional Visualization

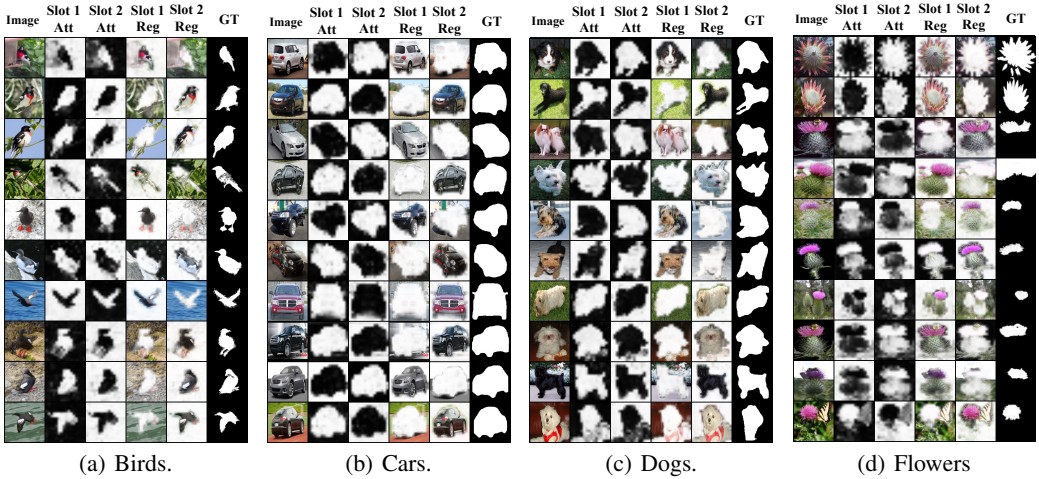

|               (a) Birds.               |               (b) Cars.               |               (c) Dogs.               |               (d) Flowers               |

Figure A2: Additional visualization of the predicted foreground masks in the unsupervised foreground extraction task on (a) Birds, (b) Cars, (c) Dogs, and (d) Flowers datasets. The design conventions follow Figure 3(a).

### A5.1  Addtional Visualization Results of the Unsupervised Foreground Extraction

| Method | FG-ARI |
|---|---|
| Slot Attention | 59.2 |
| SLATE | 61.5 |
| DINOSAUR | 64.9 |
| Invariant Slot Attention | **91.3** |
| Cyclic walks (ours) | 67.4 |

Table A3: **Results of object discovery on Clevr-Tex dataset.** We report the foreground adjusted rand index (FG-ARI) and the best is in bold.

We provide additional visualization results of the unsupervised foreground extraction task on the Birds, Cars, Dogs, and Flowers datasets in Figure A2. The same result analysis from Section 5.1 can be applied here.

### A5.2  Additional Positive and Failure Cases of the Unsupervised Object Discovery

In the unsupervised object discovery task, we provide additional positive samples of the predicted object masks in Figure A3 and negative samples in Figure A4. As discussed in Section 5.2, our method consistently predicts semantic regions despite zero annotated ground truth masks provided during training.

However, we also notice that our method as well as other methods are not perfect in some cases, especially when the number of slot bases is larger than the number of semantic objects in the scene. For example, in Row 1 of Figure A4, our method correctly segments trees, dogs, and grass, but incorrectly segments the edges of the dogs. In contrast, other methods output completely random segmented masks, carrying no semantic information whatsoever. Our method produces more "reasonable" negatively segmented masks compared with other methods, which suggests that the slot bases trained with our contrastive walks are capable of capturing distinct semantic representations and meanwhile, taking into account the holistic view of the scene.

### A5.3  Visualization Results on MOVi

We provide the visualization results of the Movi-C and Movi-E datasets in Fig A6. We found that our method predicts reasonable semantic segmentation masks in complex scenes containing multiple

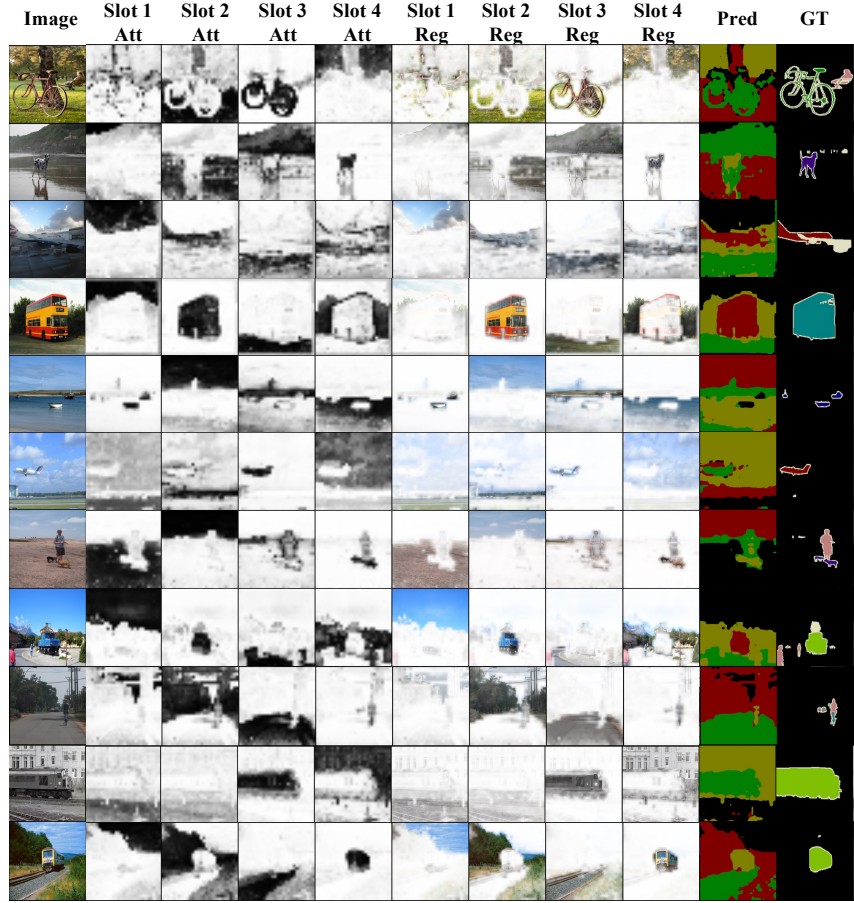

Figure A3: Additional successful samples of the predicted object masks in the unsupervised object discovery task on Pascal VOC 2012. The design conventions follow Figure 3(b).

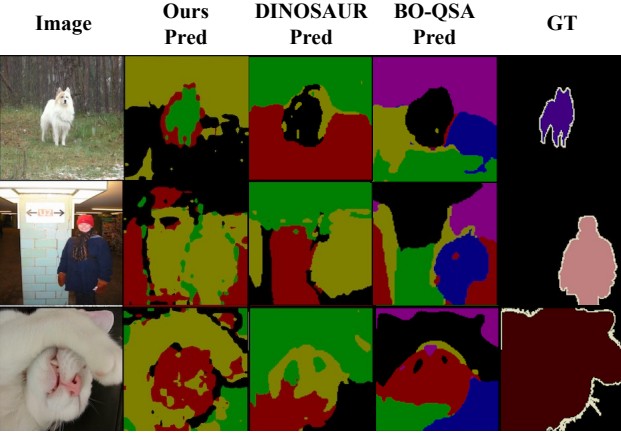

Figure A4: Failure cases of the predicted object masks in the unsupervised object discovery task on Pascal VOC 2012 dataset. From left to right, we show the original image (Col.1), the 4 predicted masks of our cyclic walks corresponding to the 4 slots (Col.2), the 4 masks for DINOSAUR (4 slots, Col.3), the 6 masks for BO-QSA (6 slots, Col.4), and the ground truth object masks (Col.5). Each color indicates a predicted mask from a slot (Col.2-4).

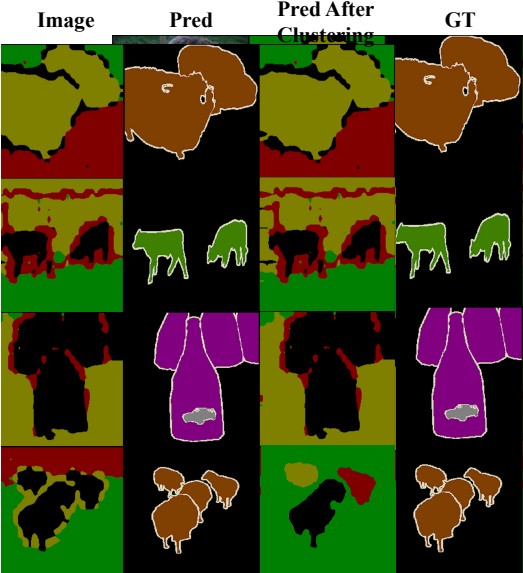

|  Image | Pred | Pred After Clustering | GT |

Figure A5: **Visualization of instance segmentation masks predicted by our cyclic walks on Pascal VOC 2012.** From left to right, we show the original image (Col.1), the predicted object instance masks (Col.2) by our default method (4 slots), the masks of our default method after applying Agglomerative Clustering [33] (Col.3, 4 clusters) and the ground truth semantic masks (Col.4). Each color indicates a predicted mask from a slot (Col.2) or a predicted mask from a cluster (Col.3).

objects (e.g. small forks are successfully segmented in Column 1, Row 3 in the cluttered scene). A similar analysis of visualization results from Fig 3(b) can be applied here for Movi-C and Movi-E.

### A5.4 Visualization Results of Instance Segmentation

Traditional object-centric representation learning methods have mainly focused on unsupervised semantic segmentation, such as DINOSAUR [37] and SlotCon [44]. Following this line of research, our method was originally designed for semantic segmentation as well. However, there is a new possibility of applying these methods in instance segmentation tasks. We made the first attempt in this direction and provided visualization results of predicted instance segmentation in Fig A5.

Inspired by CutLer [41], we extract corresponding slot representations for all the image patches and apply Agglomerative Clustering [33] to generate instance segmentation masks. Specifically, Agglomerative Clustering performs self-supervised clustering based on a distance matrix. The distance matrix takes account of both predicted object categories by our default method and the position at each pixel.

From the visualization results in Fig A5, we observed that our method produces visually reasonable instance masks after applying post-processing steps introduced in the previous paragraph. We also noticed several challenging cases where our method failed to separate object instances located close to one another (e.g. five sheep in Row 4, Fig A5). In the future, we will rigorously and quantitatively benchmark unsupervised instance segmentation models. We will also improve the designs of slot-attention modules to learn to predict instance segmentation tasks in an end-to-end fashion.

## A6 Experiments and Results Analysis on ClevrTex

We conducted experiments on the ClevrTex dataset and evaluated all baseline methods as well as our method. The results are shown in Table A3. In terms of ARI-FG, our method (67.4%) outperforms Slot-Attention (59.2%), SLATE (61.5%), and DINOSAUR (64.9%). Consistent with our results in the paper, the experimental results suggest that our method is superior at learning object-centric representations from complex scenes without reconstructions. It is worth noting that when combined

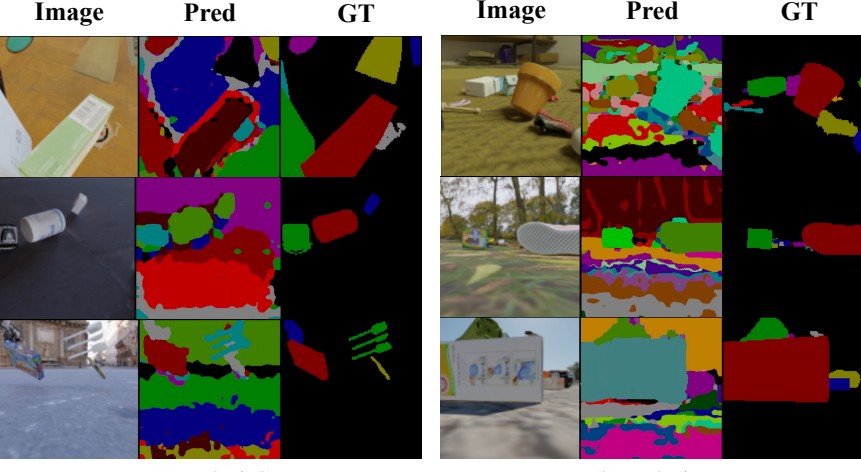

(a) MOVi-C  (b) MOVi-E

Figure A6: **Visualization of predicted semantic segmentation masks of our cyclic walks on MOVi-C and MOVi-E.** In the object discovery task, we present the predicted semantic segmentation masks on (a) MOVi-C and (b) MOVi-E. For each example, from left to right, we show the original image (Col.1), the predicted semantic segmentation masks (Col.2) of our cyclic walks (11 slots for MOVi-C and 24 slots for MOVi-E), and the ground truth semantic segmentation masks (Col.3). Each color indicates a predicted semantic segmentation mask from a slot (Col.2).

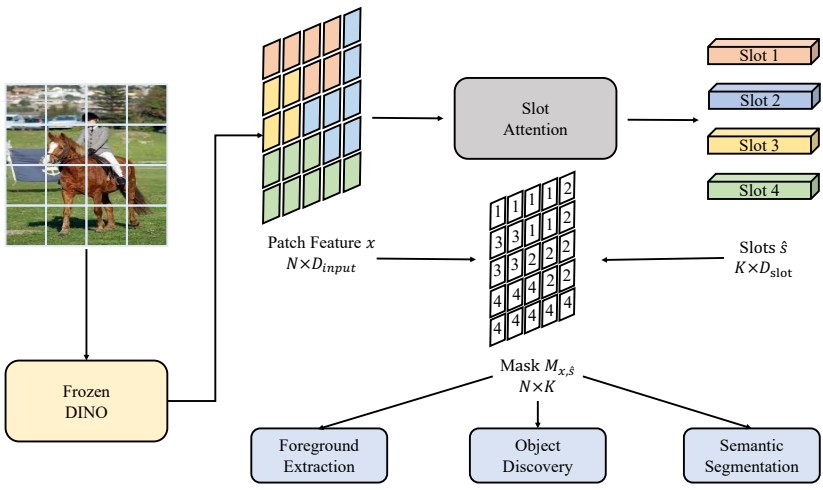

Figure A7: **Schematic illustration on how to apply the learned object-centric representations in downstream tasks.** A slot assignment mask is obtained by calculating the pairwise similarity between the features and the slot representations and then taking the corresponding slot indices with the maximum similarity score. After that, the mask can be applied to downstream tasks such as foreground extraction, object discovery, and semantic segmentation.

with a deeper CNN backbone, Slot Attention can achieve significantly higher performance: [6] reports the performance of Slot Attention on CLEVRTex to be 91.3% in FG-ARI when combined with a ResNet encoder. Compared with Slot Attention (ResNet), our method uses the frozen DINO encoder, pre-trained on naturalistic images (ImageNet). This might lead to poor feature extraction in CLEVRTex due to domain shifts between real-world and synthetic image datasets. As our experiment has shown that a good extractor is essential for our method to work well, our method can be sensitive to domain shifts if the feature extractor has not been fine-tuned on the images from the current dataset.

As shown in many existing works [29, 38] as well as our work, the original slot attention model always performs poorly. To mitigate this issue, the Invariant Slot Attention paper [6] introduces an interesting method based on slot-centric reference frames. Orthogonal to the proposed invariant slot

attention method, we introduced the cyclic walks between slots and feature maps as the supervision signals to enhance the quality of the learned slot representations. Our method does not require decoders, compared with the invariant slot method.

## A7 Limitations and Future Work

In this section, we discuss two limitations of our method. First, our model is only suitable for a fixed number of slots. When the number of slots exceeds the number of objects, our method will pay attention to unnecessary edge details. To overcome this limitation, coming up with a mechanism to merge slots or filter slots is an interesting research direction. Second, our model can not achieve instance segmentation in an end-to-end manner during the inference. How to make slots distinguish individual instances of the same category remains a challenging and practical research topic.

In addition to the future work mentioned in the main text, we would also like to look into object-centric representation learning from videos. Since videos carry temporal correlations and coherences in time, the temporal information is important for learning objects from motions. Moreover, our current work has been focusing on unsupervised foreground extraction, object discovery, and semantic segmentation tasks. In the future, we plan to explore the possibility of adapting our method and benchmarking all unsupervised object-centric learning models in high-level cognitive tasks, such as unsupervised visual question answering.

