# OpenReview forum: "Object-centric Learning with Cyclic Walks between Parts and Whole"
_NeurIPS.cc/2023/Conference — NeurIPS 2023 poster_

### Official Review · Reviewer_oB3o · 2023-07-05

**Soundness:** 3 good
**Presentation:** 2 fair
**Contribution:** 3 good
**Rating:** 6
**Confidence:** 3

**Summary:**

The paper studies the problem of unsupervised object-centric learning. Previous methods usually leverage reconstruction loss as the supervision signal. The authors propose a novel method that leverages a contrastive cyclic walk loss instead, which was originally proposed for learning correspondence between pixels in different frames of a video. The object disentanglement is achieved by using cyclic walk loss to learn the correspondence between object features and pixel features. To demonstrate the effectiveness of the proposed method, the authors conduct extensive experiments on three object-centric learning tasks. The method is also memory-efficient and computation-efficient compared with previous reconstruction-based methods.

**Strengths:**

The paper studies the problem of unsupervised object-centric learning. Previous methods usually leverage reconstruction loss as the supervision signal. The authors propose a novel method that leverages a contrastive cyclic walk loss instead, which was originally proposed for learning correspondence between pixels in different frames of a video. The object disentanglement is achieved by using cyclic walk loss to learn the correspondence between object features and pixel features. To demonstrate the effectiveness of the proposed method, the authors conducted extensive experiments on three object-centric learning tasks. The method is also memory-efficient and computation-efficient compared to previous reconstruction-based methods.

**Weaknesses:**

1. Without the reconstruction loss, the pipeline may lose most of the information of objects that are useful for downstream tasks. The paper does not conduct relevant experiments to explore this point, which is also crucial for evaluating object-centric representation learning methods. The paper would be strengthened by including experiments on predicting object properties from slot features, as in [1][2].
2. The paper lacks a discussion of the case where the number of slots is greater than the number of objects in the scene plus the background. In this case, for the whole-parts-whole cyclic walk, because the loss encourages the transition matrix to be an identity matrix, there will be some pixels that have nonzero transition probability into redundant slots. Ideally, the pixels should only transition to the slot of the corresponding object.

**Questions:**

1. It is unclear how to compute the mask from $M_{x,\hat{s}}$. Is it obtained by selecting the slot with the maximum transition probability for each pixel?
2. The paper should also contain visualizations of the model on the Movi-C and Movi-E datasets. Currently, only datasets with a fixed number of objects (1 foreground and 1 background) are shown.
3. Section 5.5 analyzes the method's efficiency in terms of model sizes, training speed, and GPU usage. What are the number of training steps for each model considered?

Minor Comments:
- The framing of Whole-Part interactions seems confusing. It's actually just interactions between image features and slot features.
- Reference [20] should be updated to the published version "Improving Object-centric Learning with Query Optimization."
- A figure explaining the whole pipeline would make the paper clearer.

**Limitations:**

No limitations were discussed. The authors conduct some failure analysis on the proposed method.

---

> ### Author Rebuttal · Authors · 2023-08-09
>
> **oB3o.1 - Weaknesses: Without the reconstruction loss, the pipeline may lose most of the information on objects that are useful for downstream tasks. The paper does not conduct relevant experiments to explore this point, which is also crucial for evaluating object-centric representation learning methods. The paper would be strengthened by including experiments on predicting object properties from slot features.**
>
> As the reviewer suggested, we now added a transformer-based decoder and the reconstruction loss to our method. In the experiment, we trained the model on Pascal VOC 2012 to investigate the effectiveness of reconstruction loss. The model with the decoder and the reconstruction loss achieved slightly better performance than our default model (from 29.6% to 29.9% in ARI-FG). This indicates that the reconstruction loss provides auxiliary information for object-centric learning. In future work, we will explore the possibilities of predicting object properties from slot features.
>
> We will include this result in Section 5.4 and a discussion about future work in Section 6.
>
> **oB3o.2 - Weaknesses: The paper lacks a discussion of the case where the number of slots is greater than the number of objects in the scene plus the background. In this case, for the whole-parts-whole cyclic walk, because the loss encourages the transition matrix to be an identity matrix, there will be some pixels that have nonzero transition probability into redundant slots. Ideally, the pixels should only transition to the slot of the corresponding object.**
>
> The reviewer correctly points out this ill-posed case when the number of slots is greater than the number of objects in the scene. Our method, as well as many other slot-based methods, is not perfect in such cases. We provided a discussion about this in lines 259-260 in the main text and visualization results of such failure cases in Supplementary Section S3, lines 37-44, and Supplementary Figure S4.
>
> **oB3o.3 - Questions: It is unclear how to compute the mask from M_{x, \hat s}. Is it obtained by selecting the slot with the maximum transition probability for each pixel?**
>
> Yes, the reviewer is right. M_{x, \hat s} is the similarity matrix between features and slots.
> It is used as the transition probability matrix from features to the slots.
>
> **oB3o.4 - Questions: The paper should also contain visualizations of the model on the Movi-C and Movi-E datasets. Currently, only datasets with a fixed number of objects (1 foreground and 1 background) are shown.**
>
> In Fig 3 in the main text, we provide visualization results on foreground extraction (1 foreground + 1 background) and object discovery (more than 3 object classes are discovered on the same image, such as ship, tree, water, and sky in Row 2 of Fig3b). As the reviewer suggested, we now provide the visualization results of the Movi-C and Movi-E datasets in Fig R2 on the rebuttal PDF. We found that our method predicts reasonable semantic segmentation masks in complex scenes containing multiple objects (e.g. small forks are successfully segmented in Column 1, Row 3 despite that the scene is cluttered).  A similar analysis of visualization results from Fig3 can be applied here for Movi-C and Movi-E (lines 255-260 in the main text).
>
> We will include the visualization results of Movi-C and Movi-E in the final version.
>
> **oB3o.5 - Questions: Section 5.5 analyzes the method's efficiency in terms of model sizes, training speed, and GPU usage. What is the number of training steps for each model considered?**
>
> We trained all models (Slot-Attention, SLATE, BO-QSA, DINOSAUR, and our Cyclic walks) with 250k training steps and selected their best models by keeping track of their best accuracies on the validation sets. Among all the models, during training, we found that our method converges the fastest. On Pascal VOC 2012 and COCO 2017, our model can achieve the best performance within 10k training steps, while other models need at least 100k training steps.
>
> We will provide this discussion in Section 5 in the final version.
>
>
> **oB3o.6 - Minor Comments: The framing of Whole-Part interactions seems confusing. It's actually just interactions between image features and slot features.**
>
> Yes, the reviewer correctly points out that whole and part refer to image features and slot features respectively. We will update these terms in the final version.
>
>
> **oB3o.7 - Minor Comments: Reference [20] should be updated to the published version "Improving Object-centric Learning with Query Optimization."**
>
> Sure. We will update the reference in the final version.
>
> **oB3o.8 - Minor Comments: A figure explaining the whole pipeline would make the paper clearer.**
>
> OK, we will add a figure describing the training process of cyclic walks and the process of applying it in various downstream tasks.

---

> > ### Comment · Reviewer_oB3o · 2023-08-15
> >
> > Thanks authors for the response. My concerns have been addressed and I will keep my positive rating.

---

> > > ### Author Response · Authors · 2023-08-15
> > > **Thanks for recommendations**
> > >
> > > We thank the reviewer for the feedback and suggestions. We will incorporate all the changes promised in the rebuttal into the final version.

---

### Official Review · Reviewer_VTwm · 2023-07-06

**Soundness:** 4 excellent
**Presentation:** 3 good
**Contribution:** 4 excellent
**Rating:** 7
**Confidence:** 4

**Summary:**

This paper introduces Cyclic Walks, an approach to obtain object-centric representations from images. The idea is to adapt contrastive random walks used for learning spatiotemporal correspondences for learning slots without a slot decoder. The other key ingredient of this approach is to use a frozen unsupervised pretrained feature extractor. The way that the approach works is that a loss is constructed based on both "whole-parts-whole" and "parts-whole-parts" cyclic walks between image patch features and slots, which trains the slot attention module used for slot extraction. The results on real world images indicate the method is effective and efficient.

**Strengths:**

#### Originality
- The main idea, to adapt contrastive random walks for training slot attention on pretrained image features, is simple and shown to be effective.
- The idea also seems like a highly novel approach for object-centric representation learning.

#### Quality

- The experiments are rigorous and cover an appropriately wide range of datasets, metrics, and baselines.

#### Clarity

- The paper is well-written and the ideas are conveyed clearly.
- Figure 1 and Figure 2 do a good job conveying the key ideas.

#### Significance

- I believe this work is a significant contribution to the unsupervised object-centric representation learning literature.

**Weaknesses:**

I have a few concerns with this work, but I believe the strengths outweigh the weaknesses.

- Missing a baseline: I would like to see Cyclic Walks compared against KMeans clustering on the frozen DINO feature tokens with $K$ equal to the number of slots used in the Cyclic Walks approach. This simple baseline would inform the extent to which the learned Slot Attention module contributes to the overall performance.
- The method introduces multiple hyperparameters, a similarity threshold and the temperature of the random walk, which appear necessary (and potentially difficult) to tune for each dataset.

**Questions:**

- Would the authors agree that using a frozen pretrained feature extractor is potentially a limitation, in that it can prevent the performance from improving beyond a point?

#### Minor suggestions for improvement

- L41-43: "First, inspired by the set encoding theory depicting that natural images can always be represented as a span of a finite set of task-dependent semantically meaningful bases". Is there a citation for this? Also, can this be re-worded into something more intuitive/less math-y? It feels out of place in the rest of this paragraph.
- L173:  CE is not defined (Cross entropy?)
- Adding a summary of the results of the ablation studies after L277 (first paragraph of Sec. 5.4) would improve readability here.

---

Update after rebuttal: Thanks to the authors for engaging with my review. I maintain my initially positive outlook on this paper.

**Limitations:**

A few failure cases are provided, but I would suggest adding a paragraph on limitations of the method to Sec. 6---space permitting. For example, discussing any hyperparameter sensitivity would be useful.

---

> ### Author Rebuttal · Authors · 2023-08-09
>
> **VTwm.1. Weaknesses: Missing a baseline: I would like to see Cyclic Walks compared against KMeans clustering on the frozen DINO feature tokens with K equal to the number of slots used in the Cyclic Walks approach. This simple baseline would inform the extent to which the learned Slot Attention module contributes to the overall performance.**
>
> As suggested by the reviewer, we now compared the performance of our method with that of direct k-means clustering (29.8% vs16.8% in ARI-FG on Pascal VOC 2012 and 39.3% vs  25.5% in ARI-FG on COCO 2017). In the experiments, our method beats k-means by a large margin, suggesting that our method is capable of learning to capture better object-centric representations.
>
> We will add these new results of k-means in Section 5.
>
> **VTwm.2. Weaknesses: The method introduces multiple hyperparameters, a similarity threshold, and the temperature of the random walk, which appear necessary (and potentially difficult) to tune for each dataset.**
>
> The hyperparameters in our method are NOT required to be tuned for individual datasets. On the contrary, in our paper, we use the SAME similarity threshold of 0.7 and the SAME temperature of 0.7 for the random walks on ALL the datasets.
>
> **VTwm.3. Questions: Would the authors agree that using a frozen pre-trained feature extractor is potentially a limitation because it can prevent the performance from improving beyond a point?**
>
> In lines 146-148, we highlighted that we follow the SAME practice as previous works [16][30] (see reference list in the main text) by freezing feature extractors. It is for a fair comparison with the baselines in the literature.
>
> We argue that freezing feature extractors is NOT a limitation of our method because of the two reasons below. First, our method can still be easily adapted to any new datasets with domain shifts. To achieve this, we introduce the 2-stage training pipeline. At stage 1, the feature extractor of a model can be trained on the new dataset using self-supervised learning. At stage 2, the feature extractor is frozen and the slot-based attention module can be fine-tuned using our method on the new dataset. Note that both stages only require self-supervised learning without any human supervision. This enables our method to be easily applied in new datasets and new domains during 2-stage training.
>
> Second, freezing feature extractors before learning object-centric representations is loosely connected with the neuroscience findings. These findings suggest that neural plasticity is increased from low-level visual processes (frozen feature extractor) to higher levels of cortical processing responsible for handling symbolic representations (slot-based attention module) [Haak et. al., 2019, Nature Communications].
>
> As suggested by the reviewer, we did the following three experiments to investigate how fine-tuning the feature extractor contributes to the overall performance of object-centric learning in ARI-FG. First, we fine-tune both the feature extractor and the slot-based attention (1-Fine-tune). The performance in ARI-FG is 12.2%. Second, we assign a small learning rate of 0.0001 for the feature extractor and a large learning rate of 0.0004 for the slot-based attention (2-Learning-rate). We observed a great performance improvement from 12.2% in 1-Fine-tune to 22.4% in 2-Learning-rate. Third, we apply EMA (Exponential Moving Averaging) on the entire model (3-EMA). The performance of 21.3% in 3-EMA is still inferior to the 2-Learning-rate.
>
> From all these results, aligning with the neuroscience findings above, we found that the slow update of the feature extractor stabilizes the learning of high-level object-centric representations. However, the performance is still inferior to our default model in the paper (29.6% in ARI-FG). This emphasizes the importance of freezing feature extractors for our method.
>
> We will include these results and discussions of our method in the final version.
>
> **VTwm.4. Minor Suggestions: L41-43: "First, inspired by the set encoding theory depicting that natural images can always be represented as a span of a finite set of task-dependent semantically meaningful bases". Is there a citation for this? Also, can this be re-worded into something more intuitive/less math-y? It feels out of place in the rest of this paragraph.**
>
> We thank the reviewer for pointing out this confusion on “set encoding theory”. This term is indeed NOT well-known. We will remove this sentence and rephrase the idea based on scene compositionality in the final version. In other words, just like DETR (https://arxiv.org/abs/2005.12872) and Mask2Former (https://arxiv.org/abs/2112.01527), these methods decompose the scene into multiple object representations or semantic masks. Similar to slot attention, these methods rely on a set of learnable “object queries”. Here, our method is to learn these “object queries” with cyclic walks in a self-supervised manner.
>
> **VTwm.5. Minor Suggestions: L173: CE is not defined (Cross entropy?)**
>
> Yes, the review is correct. CE here means Cross Entropy. We will define this in the text.
>
> **VTwm.6. Minor Suggestions: Adding a summary of the results of the ablation studies after L277 (first paragraph of Sec. 5.4) would improve readability here.**
>
> Thanks. We will add a brief summary of ablation studies, as suggested.
>
> **VTwm.7. Limitations: A few failure cases are provided, but I would suggest adding a paragraph on the limitations of the method to Sec. 6---space permitting. For example, discussing any hyperparameter sensitivity would be useful.**
>
> Yes, absolutely. We will emphasize that the hyperparameters in our method are NOT required to be tuned for individual datasets in the final version (see responses for VTwm.2). Moreover, we will expand our discussions on freezing feature extractors in Sec 6 (see responses for VTwm.3).

---

> > ### Comment · Reviewer_VTwm · 2023-08-13
> > **Thanks for the response and new experiments, some follow up questions**
> >
> > Thanks for the responses and new experiments!
> >
> > I do have some follow-up questions, however.
> >
> > ### Recommendation on hyperparameters (VTwm.6)?
> >
> > Is the recommendation of the authors then for anyone to first try 0.7 for the similarity threshold and temperature to train this method on their own data? If so, will this be mentioned in the summary of ablations (VTwm.6)?
> >
> > ### On the importance of freezing the feature extractor (VTwm.3)
> >
> > The presented results clearly indicate the importance of the two stage training approach for this method. However, the authors write in the discussion (L337-340) "our method has been developed on a frozen unsupervised feature extractor. In the future, hierarchical contrastive walks can be explored in any feed-forward architectures, where the models can **simultaneously** learn both pixel-level and object-centric representations incrementally over multiple layers" (emphasis mine).
> >
> > Can the authors clarify for me why they suggest exploring training an architecture with cyclic walks without freezing the feature extractor here? This seems contradictory to the presented results that show the importance of freezing the feature extractor.
> >
> > Are there any other key future directions?

---

> > > ### Author Response · Authors · 2023-08-14
> > > **Response to follow-up questions**
> > >
> > > **VTwm.F1. Questions: Is the recommendation of the authors for anyone to first try 0.7 for the similarity threshold and temperature to train this method on their own data? If so, will this be mentioned in the summary of ablations (VTwm.6)?**
> > >
> > > First, we apologize for a typo in the response of VTwm.2. Rather than 0.7, we used the temperature of **0.1**, as also indicated in line 6 in our Supp. Material.
> > >
> > > In the final version, we will clarify in the summary of ablations (line 277) that we encourage readers to use a similarity threshold of 0.7 and a temperature of 0.1 for their own data, just like what we did over all the datasets from all the tasks.
> > >
> > > We will also add the following statement in line 277: “From the ablation study below, we have found that these two hyperparameter values are relatively robust to various datasets. See the ablation studies below where we provide the variations on hyperparameter choices to examine their sensitivity in the performance.”
> > >
> > >
> > > **VTwm.F2. Questions: The presented results clearly indicate the importance of the two-stage training approach for this method. However, the authors write in the discussion (L337-340) "our method has been developed on a frozen unsupervised feature extractor. In the future, hierarchical contrastive walks can be explored in any feed-forward architectures, where the models can simultaneously learn both pixel-level and object-centric representations incrementally over multiple layers" (emphasis mine).
> > > Can the authors clarify for me why they suggest exploring training an architecture with cyclic walks without freezing the feature extractor here? This seems contradictory to the presented results that show the importance of freezing the feature extractor.
> > > Are there any other key future directions?**
> > >
> > > We respectfully disagree with the reviewer that what we commented about two-stage training in the rebuttal is contradictory to the future work mentioned in the main text.
> > >
> > > For example, a model which can detect objects but cannot segment images does not imply that the model has LIMITATION/DRAWBACK of not being able to perform image segmentation. Similarly, our current model which can learn good object-centric representations but cannot learn pixel-level representations does not imply that our model has limitations in object-centric representation learning. Instead, we found that good pixel-level representations facilitate object-centric representation learning in our method (see our original response in VTwm.3).
> > >
> > > Of course, in the real world, we want a general AI model which can ideally do all the tasks all at once. This includes a model which can learn both pixel-level and object-centric representations at the same time. Together with the entire research community, we are excited about working on this problem in the future, i.e. designing an AI model which can simultaneously learn both pixel-level and object-centric representations.
> > >
> > > In addition to the future work mentioned in the main text, we would also like to look into object-centric representation learning from videos, as videos carry temporal correlations and coherences, important for learning objects from motions. Moreover, our current work has been focusing on unsupervised foreground extraction, object discovery, and semantic segmentation tasks. In the future, we plan to explore the possibility of adapting our method and benchmarking all unsupervised object-centric learning models on unsupervised instance segmentation tasks and unsupervised visual question answering.
> > >
> > > In the final version, we will clarify the future work on learning pixel-level and object-centric representations simultaneously and provide the additional future directions discussed here in Section 6.

---

> > > > ### Comment · Reviewer_VTwm · 2023-08-14
> > > > **No further questions**
> > > >
> > > > Thanks to the authors for taking the time to answer my follow up questions

---

> > > > > ### Author Response · Authors · 2023-08-15
> > > > > **Thanks for recommendations**
> > > > >
> > > > > We thank the reviewer for the feedback and suggestions. We will incorporate all the changes promised in the rebuttal into the final version.

---

### Official Review · Reviewer_8Kba · 2023-07-06

**Soundness:** 3 good
**Presentation:** 3 good
**Contribution:** 3 good
**Rating:** 6
**Confidence:** 4

**Summary:**

This paper works on unsupervised object discovery, that is, learning to decompose the compositional components of a scene. Based on frozen DINO features, it proposes to introduce random walks between part-level features (dense output of DINO) and object-level features (output of SlotAttention). Each object-level node is encouraged to go back to the same node after random walking to and back from local features, and vice-versa for each local node. Extensive experiments validate the effectiveness of the proposed pipeline.

**Strengths:**

*Originality*: Tough SlotAttention module is not novel, and cycle consistency has been adopted in some previous works. These works are well acknowledged in the text, and the contribution from this paper: constructing cycle consistency between part-level and object-level features as a supervisory signal for object discovery with SlotAttention is absolutely original. The idea is clean and shown to be effective.

*Quality*: This work is relatively professional and of good quality. The proposed method is clean and reasonable, and has shown effectiveness in multiple benchmarks. Besides, error bars are provided and make the results more reliable. I also would like to note that many previous works did not mention STEGO in their tables, and I am happy to see it clearly compared in this work.

*Clarity*: The delivery is fairly smooth and clear, and I find no major issues in understanding this paper.

*Significance*: I am happy to see another reconstruction-free framework being validated on real-world object discovery, which is rare in this community. I also noticed in Tab.3 that this work is robust to pre-training feature extractors, which deviates from the recent trend of ''DINO is all you need'', and is good for the community.

*Reproducibility*: Code is not provided but the text is relatively clear.

**Weaknesses:**

Kindly suggest for direct comparison with the following related works:

- [MaskDistill] W. Van Gansbeke et al., Discovering Object Masks with Transformers for Unsupervised Semantic Segmentation, arXiv 2022.
- [Odin] O. J. Hénaff et al., Object Discovery and Representation Networks, ECCV 2022.
- [SlotCon] X. Wen et al., Self-Supervised Visual Representation Learning with Semantic Grouping, NeurIPS 2022.
- [COMUS] A. Zadaianchuk et al., Unsupervised Semantic Segmentation with Self-Supervised Object-Centric Representations, ICLR 2023.

**Questions:**

- Regarding the efficiency issue, what is the inference speed, and how is it compared with other frameworks?
- In the abstract it is claimed to be capable of CNN features, but the results are all with ViTs. Could the effectiveness with CNN features be validated?

Minor:

- L76: missing space
- L108: slot attention does not directly adopt cross-attention, the dimension for normalizing the attention weights is different and thus to introduce competition between slots
- Fig.4: please polish this figure for clarity
- L321: twice fewer is not clear

**Limitations:**

limitations are not explicitly discussed, but failure cases are provided in the appendix

---

> ### Author Rebuttal · Authors · 2023-08-09
>
> **8Kba.1 - Weaknesses: Kindly suggest for direct comparison with the following related works.**
>
> We now include the following results of all the methods suggested by the reviewer on Pascal VOC 2012 and COCO-stuff27 in the table below. We are unable to compare with Odin, since Odin is a self-supervised instance segmentation method. Our method gets competitive performances compared to these methods, suggesting that our method is capable of learning to capture reasonable object-centric representations.
>
> We will cite and discuss these works provided by the reviewer in Section 2. and include these new results of the baselines in Section 5.
> |            mIoU          |        ours     |  MaskDistill  |       Odin      |      SlotCon   |    COMUS    |
> |:------------------------:|:---------------:|:----------------:|:----------------:|:----------------:|:----------------:|
> | Pascal VOC 2012 |        43.3     |       42.0       |        N.A.      |         N.A.      |        50.0      |
> |    COCO-stuff27    |        22.5     |       N.A.       |        N.A.      |         18.3      |        N.A.      |
>
> **8Kba.2 - Questions: Regarding the efficiency issue, what is the inference speed, and how is it compared with other frameworks?**
>
> For fair comparisons, we benchmark the inference speed of all the methods using the same hardware and data configurations: (1) one single RTX-A5000 GPU; and (2) input image size of 224 × 224, batch size of 8, and 4 slots. We provide the inference speed in the table below. Among all the methods, our cyclic walk is the most efficient with the fastest inference speed.
>
> In the final version, we will include these results in Section 5.
> |           infer. speed (img/s)          |        ours     |         SA       |      SLATE    |  DINOSAUR |   BOQ-SA    |
> |:-----------------------------------------:|:---------------:|:----------------:|:----------------:|:----------------:|:----------------:|
> |           Object Discovery             |        285      |         126      |          32       |          130      |          32       |
>
> **8Kba.3 - Questions: In the abstract, it is claimed to be capable of CNN features, but the results are all with ViTs. Could the effectiveness of CNN features be validated?**
>
> We thank the reviewer for pointing this out. Here, we replace DINO with ResNet50 to extract features as the “whole” and perform our cyclic walks between these features and the slots. Our method with ResNet achieves similar performance as our method with DINO in ARI-FG on Pascal VOC 2012 (28.3% vs 29.7%). We will include this result in our final version.
>
> **8Kba.4 - Minor**
>
> Thank you! We will fix the text, rephrase the sentences, and polish the figure.

---

> > ### Comment · Reviewer_8Kba · 2023-08-15
> >
> > Thanks to the authors for the response and I have no further concerns. I feel okay that it performs worse than COMUS since the method is clean and is an orthogonal effort. I do agree that whether DINO features are used should be clearly marked in comparisons since it is a key factor in this field. I confirm my positive recommendation.

---

> > > ### Author Response · Authors · 2023-08-15
> > > **Thanks for recommendations**
> > >
> > > We thank the reviewer for the feedback and suggestions. We will incorporate all the changes promised in the rebuttal into the final version.

---

### Official Review · Reviewer_38HN · 2023-07-14

**Soundness:** 3 good
**Presentation:** 3 good
**Contribution:** 2 fair
**Rating:** 5
**Confidence:** 4

**Summary:**

The paper presents a method for unsupervised object discovery, while using cyclic walks between part and whole features as a supervision signal. While previous methods mainly use RGB or feature reconstruction as supervisory signal, this method uses a form of contrastive learning.  Their lack of decoder architecture due to contrastive learning, reduces the overall computational overhead from the previous reconstruction based models that explicitly required a decoder. They showcase their result on seven image datasets and on three unsupervised tasks. They compare against many baselines and indicate better object discovery performance than most of the baselines.

**Strengths:**

i) Interesting and novel idea of using random cyclic walks as supervisory signal.

ii) Method is well written and easy to understand.

iii) Good performance on real world datasets for object discovery.

**Weaknesses:**

i) The results on classical object centric datasets such as CLEVR or CLEVRTex are missing,

ii) Missing results for just training on RGB pixels, this would be interesting to see even if the results are not good, it can help get better intutions of where the method works/fails

iii) Missing baseliens such as CutLer(https://arxiv.org/abs/2301.11320) or SlotCon (https://github.com/CVMI-Lab/SlotCon) or simply doing kmeans clustering on the Dino features.

iv) Does the model get instance segmentation or semantic segmentation. Didn't see any visuals that indicate the model can learn instance segmentation.

**Questions:**

i) How does the method compare against reconstruction methods on original CLEVR or ClevrTex dataset, any one is fine?

ii) How does the method do when training on top of RGB pixels instead of pre-trained features.

iii) How does the method compare against SlotCon or Cutler or simple kmeans on features?

iv) Visuals/results indicating this is capable of learning instance segments?

**Limitations:**

Yes

---

> ### Author Rebuttal · Authors · 2023-08-09
>
> **38HN.1 - Questions: How does the method compare against reconstruction methods on the original CLEVR or ClevrTex dataset, is anyone fine?**
>
> As suggested by the reviewer, we conducted experiments on the ClevrTex dataset and evaluated the performance of all baseline methods and our method. In terms of ARI-FG, our method (67.4%) outperforms Slot-Attention (59.2%), SLATE (61.5%), and DINOSAUR (64.9%). Consistent with our results in the paper, the experimental results suggest that our method is superior at learning object-centric representations from complex scenes without reconstructions.
>
> We will add these results and discussions in the final version.
>
> **38HN.2 - Questions: How does the method do when training on top of RGB pixels instead of pre-trained features?**
>
> In lines 146-148, we highlighted that we follow the SAME practice as previous works [16][30] (see reference list in the main text) by freezing feature extractors. It is for a fair comparison with the baselines in the literature.
>
> We argue that freezing feature extractors is NOT a limitation of our method because of the two reasons below. First, our method can still be easily adapted to any new datasets with domain shifts. To achieve this, we introduce the 2-stage training pipeline. At stage 1, the feature extractor of a model can be trained on the new dataset using self-supervised learning. At stage 2, the feature extractor is frozen and the slot-based attention module can be fine-tuned using our method on the new dataset. Note that both stages only require self-supervised learning without any human supervision. This enables our method to be easily applied in new datasets and new domains during 2-stage training.
>
> Second, freezing feature extractors before learning object-centric representations is loosely connected with the neuroscience findings. These findings suggest that neural plasticity is increased from low-level visual processes (frozen feature extractor) to higher levels of cortical processing responsible for handling symbolic representations (slot-based attention module) [Haak et. al., 2019, Nature Communications].
>
> As suggested by the reviewer, we did the following three experiments to investigate how fine-tuning the feature extractor contributes to the overall performance of object-centric learning in ARI-FG. First, we fine-tune both the feature extractor and the slot-based attention (1-Fine-tune). The performance in ARI-FG is 12.2%. Second, we assign a small learning rate of 0.0001 for the feature extractor and a large learning rate of 0.0004 for the slot-based attention (2-Learning-rate). We observed a great performance improvement from 12.2% in 1-Fine-tune to 22.4% in 2-Learning-rate. Third, we apply EMA (Exponential Moving Averaging) on the entire model (3-EMA). The performance of 21.3% in 3-EMA is still inferior to the 2-Learning-rate.
>
> From all these results, aligning with the neuroscience findings above, we found that the slow update of the feature extractor stabilizes the learning of high-level object-centric representations. However, the performance is still inferior to our default model in the paper (29.6% in ARI-FG). This emphasizes the importance of freezing feature extractors for our method.
>
> We will include these results and discussions of our method in the final version.
>
> **38HN.3 - Questions: How does the method compare against SlotCon, CutLer, or simple k-means on features?**
>
> We now include the following results of all the methods suggested by the reviewer. Since Cutler is designed for instance segmentation; we excluded it from the method comparisons in the unsupervised semantic segmentation task. We compared our method with SlotCon (22.5% vs 18.3% in mIoU on COCO-stuff27). In addition, we compared the performance of our method with that of direct K-means clustering (29.8% vs16.8% in ARI-FG on Pascal VOC 2012 and 39.3% vs  25.5% in ARI-FG on COCO 2017). In the experiments, our method beats SlotCon and K-means by a large margin, suggesting that our method is capable of learning to capture better object-centric representations.
>
> We will cite and discuss these works provided by the reviewer in Section 2, and add these new results of the baselines in Section 5.
>
> **38HN.4 - Questions: Visuals/results indicating this is capable of learning instance segments?**
>
> Traditional object-centric representation learning methods have been mainly focusing on unsupervised semantic segmentation, such as DINOSAUR and SlotCon (see reference list in the main paper). Following this line of research, our method was originally designed for semantic segmentation as well.
>
> It is interesting that the reviewer highlighted the new possibility of applying these methods in instance segmentation tasks. Inspired by CutLer, we extract corresponding slot representations for all the image patches and apply Agglomerative Clustering [scikit-learn library] to generate instance segmentation masks. Specifically, Agglomerative Clustering performs self-supervised clustering based on a distance matrix. The distance matrix takes account of both predicted object categories by our default method and the position at each pixel. We provide visualization results of predicted instance segmentation in Fig R1 on the rebuttal PDF page.
>
> From the visualization results, we observed that our method produces visually reasonable instance masks after applying post-processing steps. We also noticed several challenging cases where our method fails to separate object instances located close to one another (e.g. five sheep in Row 4). In the future, we will rigorously and quantitively benchmark unsupervised instance segmentation models. We will also improve the designs of slot-attention modules to learn to predict instance segmentation tasks in an end-to-end fashion.
>
> In the final version, we will include discussions on future works of unsupervised instance segmentation in Section 6.

---

> > ### Comment · Reviewer_38HN · 2023-08-14
> >
> > Thanks for the additional results.
> >
> > Q) "Why does the method perform worse than COMUS ?":
> >
> > The performance of the method compared to COMUS seems significantly worse in response to reviewer 8Kba, that is 50.0 vs 43.3. Does this hold true across all benchmarks? Can the authors perform dense evaluation of COMUS across all benchmark? If not can the authors justify as to why one would use their method over COMUS?
> >
> > Q) Lack of comparisions on without using pre-trained features:
> >
> > To the best of my knowledge , Amongst all the methods the paper compares against only DINOSAUR uses pre-trained DINO features, this makes comparisions against most baselines a bit unfair. I would recommend highlighting this in the Table.  Additionaly i would expect a lot dense comparision wrt Dinosaur specifically on the datasets they report in their paper. I was unable to find a result for Bird, Dogs etc in their paper. I think comparing with DINOSAUR , while using the same numbers from their paper would make the current results a lot stronger.

---

> > > ### Author Response · Authors · 2023-08-15
> > > **Thanks for comments. Response to follow-up questions**
> > >
> > > **38HN.F1 - Questions: Why does the method perform worse than COMUS?**
> > >
> > > As the reviewer correctly points out, our method underperforms COMUS in mIoU metrics on Pascal VOC 2012(43.3% versus 50.0%).
> > >
> > > However, as also agreed by Reviewer 8Kba, we argue that, in comparison to our method, there are two additional components in COMUS possibly attributing to the performance differences and unfair comparisons. First, COMUS employs two pre-trained saliency detection architectures DeepUSPS (Nguyen et al., 2019, NeurIPS, https://arxiv.org/abs/1909.13055) and BasNet (Qin et al., 2019, CVPR, https://arxiv.org/abs/2101.04704) in addition to DINO. The saliency detection model requires an additional MSRA image dataset (Cheng et al., 2015, TPAMI, https://arxiv.org/abs/2101.04704) during pre-training. Thus, compared to our cyclic walks,  COMUS indirectly relies on extra useful information. Second, COMUS uses Deeplab-v3 as its backbone for predicting semantic segmentation masks. For better segmentation performances, Deeplab-v3 extracts large-resolution feature maps from images and applies Atrous Spatial Pyramid Pooling for aggregating these feature maps over multiple scales. In contrast, our method extracts low-resolution feature maps with DINO and is not specifically designed for unsupervised semantic segmentation. Yet, it is remarkable that our method still achieves comparable performances as COMUS in unsupervised semantic segmentation tasks.
> > >
> > > In addition to the comparable performance with COMUS in unsupervised semantic object segmentation, our method is capable of parsing the entire scene on the image into semantically meaningful regions, which COMUS fails to do. These include distinct backgrounds, such as sky, lake, trees, and grass. For example, in Fig 3(b) in the main body, our method successfully segments backgrounds, such as the lake, the trees, and the sky. However, COMUS fails to segment background elements. For example, in Column 4 of Fig 1 in the paper (https://arxiv.org/abs/2207.05027), COMUS only segments the foreground ships and fails to segment the lake and the sky. Segmenting both salient objects and other semantic regions is essential for many computer vision applications. This emphasizes the importance and unique advantages of our method.
> > >
> > > In the final version, we will highlight the differences between COMUS and our method and discuss the advantages of our method over COMUS in Section 2.
> > >
> > > **38HN.F2 - Questions:Lack of comparisions on without using pre-trained features.**
> > >
> > > We thank the reviewer for the recommendation. In the final version, we will highlight that both DINOSAUR and our method use the pre-trained frozen DINO in Table 1.
> > >
> > > Note that the results of DINOSAUR reported in our paper deviate slightly from the results reported in the original DINOSAUR paper (Seitzer et al., 2023, ICLR, https://arxiv.org/abs/2209.14860). This is because the code for DINOSAUR was not available before the deadline for Neurips submission. Hence, we strictly followed the model specifications in the original paper, re-implemented DINOSAUR on our own, and reported the results of our re-implemented version. These results include the performances of our re-implemented DINOSAUR on Birds, Dogs, Cars, and Flowers datasets, which were missing in the original DINOSAUR paper. We will release our re-implementation code of all the baselines upon publication.
> > >
> > > As the reviewer pointed out, in the table below, we now copied the exact same results in the original DINOSAUR paper. For easy comparison, we also copied the results of our re-implemented DINOSAUR and our method from our paper. The ARI-FG metric performances in (mean+/- standard deviation) are reported.
> > >
> > > From the results, we observed that our method outperformed the re-implemented DINOSAUR and performed competitively well as the original DINOSAUR in all the experiments. Note that in comparison to DINOSAUR, our method does not require decoders for reconstruction and hence, our method has fewer parameters, less GPU memory usage, converges faster during training, and achieves faster inference speed as indicated in Section 5.5.
> > >
> > > In the final version, we will include this table, dedicate a short paragraph to highlight the differences between our method and DINOSAUR and discuss the advantages of our method over DINOSAUR.
> > > |           |Pascal VOC 2012|COCO 2017| MOVi-C|MOVi-E|
> > > |:-------:|:----------------------:|:----------------:|:---------:|:---------:|
> > > |Original DINOSAUR|24.6±0.2|40.5±0.0|67.2±0.3|64.7±0.7|
> > > |Re-implemented DINOSAUR|27.5±0.2|36.2±0.7|64.0±0.5|62.4±0.7|
> > > |Our method|29.6±0.8|39.7±0.8|67.6±0.3|64.7±0.7|

---

> > ### Comment · Area_Chair_ht8A · 2023-08-17
> > **CLEVRTex Slot Attention baseline**
> >
> > Dear authors,
> >
> > Thank you for adding an experiment on CLEVRTex. Please note that Slot Attention, when combined with a deeper CNN backbone, can achieve significantly higher performance: the Invariant Slot Attention paper [Biza et al., 2023] reports a performance of Slot Attention on CLEVRTex of 91.3% FG-ARI when combined with a ResNet encoder. I think that the experiment nonetheless provides value, but I would recommend discussing it in the context of more recent results (in this case, a result from February 2023).
> >
> > Thank you.
> >
> > --Your AC

---

> > > ### Author Response · Authors · 2023-08-18
> > > **Thanks for AC's advice**
> > >
> > > **AC: Thank you for adding an experiment on CLEVRTex. Please note that Slot Attention, when combined with a deeper CNN backbone, can achieve significantly higher performance: the Invariant Slot Attention paper [Biza et al., 2023] reports the performance of Slot Attention on CLEVRTex of 91.3% FG-ARI when combined with a ResNet encoder. I think that the experiment nonetheless provides value, but I would recommend discussing it in the context of more recent results (in this case, a result from February 2023).**
> > >
> > > We appreciate the advice from AC. In the final version, we will incorporate the result of SA(ResNet) and provide the discussion below in the result analysis:
> > >
> > > Compared with SA (ResNet) in the paper, our method uses the frozen DINO encoder, pre-trained on naturalistic images (ImageNet). This might lead to poor feature extraction in CLEVRTex due to domain shifts between real-world and synthetic image datasets. As our experiment has shown that a good extractor is essential for our method to work well (see the response of **VTwm.3**), our method can be sensitive to domain shifts if the feature extractor has not been fine-tuned on the images from the current dataset.
> > >
> > > The original slot attention model does not always perform well as shown in the Invariant Slot Attention paper, our work, as well as many others. To mitigate this issue, the Invariant Slot Attention paper introduces an interesting method based on slot-centric reference frames. Orthogonal to the proposed invariant slot attention method, we introduced the cyclic walks between slots and feature maps as the supervision signals to enhance the quality of the learned slot representations. Our method does not require decoders, compared with the invariant slot method.
> > >
> > > We will cite the Invariant Slot Attention paper and discuss its differences with our work in Section 2 in the revised version.

---

> > > > ### Comment · Reviewer_38HN · 2023-08-18
> > > >
> > > > Thanks for the information.
> > > >
> > > > I think the method primarily works for semantic segmentation and that's why the results on ClevrTex are poor compared to SlotAttention. I do think the authors should make this clear in their paper or try to beat the 91.3 score of SA on ClevrTex.  For semantic segmentation the work achieves comparable results to dinosaur, while not having a decoder. Mainly due to the distinctiveness of the approach and comparable results with DINOSAUR i have increased my rating to borderline accept.

---

> > > > > ### Author Response · Authors · 2023-08-18
> > > > > **Thanks for comments**
> > > > >
> > > > > We thank the reviewer for the feedback. We will incorporate this feedback in the revised version.

---

### Author Rebuttal · Authors · 2023-08-09

We thank the reviewers for feedbacks and suggestions. We present the figures in PDF and the responses to individual reviewers’ questions below. The original questions from the reviewers are copied and bolded.

---

### Decision · Program_Chairs · 2023-09-21

**Decision:**

Accept (poster)

**Comment:**

I agree with the reviewers that this is an interesting and novel contribution to the emergent area of unsupervised object-centric learning. The paper is well-written and of high quality. The reviewers especially appreciated that the method does not rely on feature reconstruction, which is rare in this area and any progress in this direction should be of interest to the community.

While the method achieves similar unsupervised segmentation performance as recent related work (DINOSAUR), it explores an alternative learning approach via contrastive learning. Compared to established reconstruction-based approaches such as DINOSAUR, the proposed method does not require a decoder and therefore achieves higher inference throughput, which is a valuable contribution.

I agree with the reviewers that the paper meets the bar for acceptance, but it would benefit from a careful revision that incorporates the additional experiments and feedback from the review period (echoing also the conclusions of reviewer 38HN). I think that the rebuttal contains sufficient clarifications and information that necessary changes/edits can be done in time for the camera-ready deadline without requiring further review.

In particular – in addition to incorporating the reviewer feedback – please discuss the new results on CLEVRTex (especially the gap to the Slot Attention baseline) as well as the gap to the COMUS baseline on Pascal VOC in detail. It would be valuable for the community to understand why the proposed methods underperforms in these two comparisons.